# Heterogeneity of the GFP fitness landscape and data-driven protein design

Louisa Gonzalez Somermeyer[1], Aubin Fleiss[2,3], Alexander S Mishin[4],
Nina G Bozhanova[5], Anna A Igolkina[6], Jens Meiler[5,7],
Maria-Elisenda Alaball Pujol[2,3], Ekaterina V Putintseva[8], Karen S Sarkisyan[2,3,4]*,
Fyodor A Kondrashov[1,9]*

[1]Institute of Science and Technology Austria, Klosterneuburg, Austria; [2]Synthetic Biology Group, MRC London Institute of Medical Sciences, London, United Kingdom; [3]Institute of Clinical Sciences, Faculty of Medicine and Imperial College Centre for Synthetic Biology, Imperial College London, London, United Kingdom; [4]Shemyakin-Ovchinnikov Institute of Bioorganic Chemistry, Russian Academy of Sciences, Moscow, Russian Federation; [5]Department of Chemistry, Center for Structural Biology, Vanderbilt University, Nashville, United States; [6]Gregor Mendel Institute, Austrian Academy of Sciences, Vienna BioCenter, Vienna, Austria; [7]Institute for Drug Discovery, Medical School, Leipzig University, Leipzig, Germany; [8]LabGenius, London, United Kingdom; [9]Evolutionary and Synthetic Biology Unit, Okinawa Institute of Science and Technology Graduate University, Okinawa, Japan

*For correspondence:
karen.s.sarkisyan@gmail.com
(KSS);
fyodor.kondrashov@oist.jp (FAK)

Competing interest: The authors declare that no competing interests exist.

**Abstract** Studies of protein fitness landscapes reveal biophysical constraints guiding protein evolution and empower prediction of functional proteins. However, generalisation of these findings is limited due to scarceness of systematic data on fitness landscapes of proteins with a defined evolutionary relationship. We characterized the fitness peaks of four orthologous fluorescent proteins with a broad range of sequence divergence. While two of the four studied fitness peaks were sharp, the other two were considerably flatter, being almost entirely free of epistatic interactions. Mutationally robust proteins, characterized by a flat fitness peak, were not optimal templates for machine-learning-driven protein design – instead, predictions were more accurate for fragile proteins with epistatic landscapes. Our work paves insights for practical application of fitness landscape heterogeneity in protein engineering.

## Editor's evaluation

Using high throughput mutagenesis, this work shows that evolutionary distance between homologous genes is not predictive of how these genes' functions will change in response to similar mutations. This suggests that the starting gene sequence will influence how the synthetic design of new protein functions can occur and also supports a role for conditionality in the natural evolution of protein functions.

## Introduction

Understanding the relationship between genotype and phenotype, the fitness landscape, elucidates the fundamental laws of heredity (*Canale et al., 2018*; *de Visser and Krug, 2014*; *Ferretti et al., 2018*; *Fragata et al., 2019*; *Wright, 1932*) and may ultimately create novel methods of protein design (*Alley et al., 2019*; *Bryant et al., 2021*; *Hirabayashi and Arai, 2019*; *Wrenbeck et al., 2017*; *Wu et al., 2019*). The fitness landscape is often conceptualised as a multidimensional surface (*de Visser*

*and Krug, 2014*; *Ferretti et al., 2018*; *Kondrashov and Kondrashov, 2015*; *Wright, 1932*) with one dimension representing fitness, or another phenotype, and the other dimensions each representing a genotype's locus. Originally, the fitness landscape was introduced to describe the relationship between fitness and the entire genome (*de Visser and Krug, 2014*; *Wright, 1932*). Over time, the usefulness of the concept of the fitness landscape led to the adaptation of this term to describe the relationship between protein function and its protein-coding gene sequence (*Biswas et al., 2021*; *Ogden et al., 2019*; *Romero and Arnold, 2009*; *Wittmann et al., 2021*; *Zheng et al., 2020*). Absolute knowledge of the fitness landscape would reveal the phenotypes conferred by any arbitrary genotype (*de Visser and Krug, 2014*; *Ferretti et al., 2018*; *Fragata et al., 2019*), with immense and obvious practical implications (*Alley et al., 2019*; *Bryant et al., 2021*; *Hirabayashi and Arai, 2019*; *Kemble et al., 2019*; *Wrenbeck et al., 2017*; *Wu et al., 2019*). However, sparse experimental data, even for specific genes, and the concomitant lack of understanding of the rules by which fitness landscapes are formed, limit the accuracy of phenotype predictions based on sequence alone (*Lässig et al., 2017*) but see *Bryant et al., 2021*; *Rocklin et al., 2017*; *Senior et al., 2020*; *Wu et al., 2019*.

While several experimentally characterized fitness landscapes for specific proteins have been reported (*Hartman and Tullman-Ercek, 2019*; *Jacquier et al., 2013*; *Kuo et al., 2020*; *Melamed et al., 2013*; *Olson et al., 2014*; *Sarkisyan et al., 2016*), such surveys of large proteins are still hindered by the enormity of the genotype space (*de Visser and Krug, 2014*; *Wright, 1932*). Even for the Green Fluorescent Protein (GFP), which is only ~250 amino acids long, there are $20^{250}$ possible genotypes. Without complex epistatic interactions between amino acid sites the fitness landscape could be deduced from the independent contribution of each amino acid at each site (*Kondrashov and Kondrashov, 2015*), requiring just 5000 (20*250) measurements of the effects of all single mutations in GFP. However, epistatic interactions between amino acid sites are common (*Russ et al., 2020*) and many of them are too complex to predict with available data (*Pokusaeva et al., 2019*). Despite some advances in the development of data-driven approaches to protein design (*Biswas et al., 2021*; *Biswas et al., 2018*; *Bryant et al., 2021*; *Kemble et al., 2019*), it is still not clear what fraction of the $20^{250}$ sequences of the GFP, or any other gene, must be characterized to approach the coveted absolute knowledge of the fitness landscape (*Kemble et al., 2019*; *Sailer et al., 2020*; *Zhou and McCandlish, 2020*).

Despite lack of data, experiments and theory provide some insights on the global fitness landscape (*Fragata et al., 2019*; *Kemble et al., 2019*). Each extant genotype, one that is found in an extant species, is a point of high fitness, or a fitness peak, on the highly dimensional and extraordinarily large genotype space (*de Visser and Krug, 2014*; *Fragata et al., 2019*; *Smith, 1970*; *Wright, 1932*). These extant genotypes had a common ancestor, so they must be connected by ridges of high fitness (*Gong et al., 2013*; *Smith, 1970*; *Povolotskaya and Kondrashov, 2010*). Nevertheless, only an infinitesimally small fraction of all genotypes are functional (fewer than $10^{-11}$), those that correspond to fitness peaks and ridges, and the remaining genotypes confer low fitness (*Keefe and Szostak, 2001*). The fitness peaks are sharp (*Bank et al., 2015*; *Melamed et al., 2013*; *Sarkisyan et al., 2016*) and the ridges are narrow (*Gong et al., 2013*; *Kumar et al., 2017*; *Pokusaeva et al., 2019*; *Sailer et al., 2020*) and, on average, only a few random mutations in a wildtype sequence reduce its fitness to zero (*Hartman and Tullman-Ercek, 2019*; *Kemble et al., 2019*). The sharpness of the peaks is enhanced by negative epistasis, such that a genotype with several random mutations has a lower fitness than expected if mutations acted independently (*Haddox et al., 2018*; *Sarkisyan et al., 2016*). Thus, a random walk from a fitness peak eventually leads to an area of the genotype space where only an infinitesimally small fraction of sequences are functional, likely explaining why accurate prediction of functional genotypes at a substantial distance away from a functional genotype remains a challenge (*Alley et al., 2019*; *Hirabayashi and Arai, 2019*; *Russ et al., 2020*; *Wu et al., 2019*).

The shape of fitness peaks and ridges, and their distribution in genotype space has implications for fundamental questions in evolution (*de Visser and Krug, 2014*) and practical applications (*Sardanyés et al., 2008*). Evolution starting at a sharp fitness peak is expected to proceed at a different pace than evolution on a flat one (*Bershtein et al., 2006*; *Codoñer et al., 2006*; *de Visser et al., 2003*; *Draghi et al., 2010*; *Wagner, 2008*). Furthermore, it has been suggested that flat fitness peaks, representing robust genotypes, may be evolutionarily preferable to sharp peaks, which represent fragile genotypes (*Bershtein et al., 2006*; *de Visser et al., 2003*; *Draghi et al., 2010*; *Klug et al., 2019*; *Zheng et al., 2020*). However, how different shapes of fitness peaks may be distributed in genotype

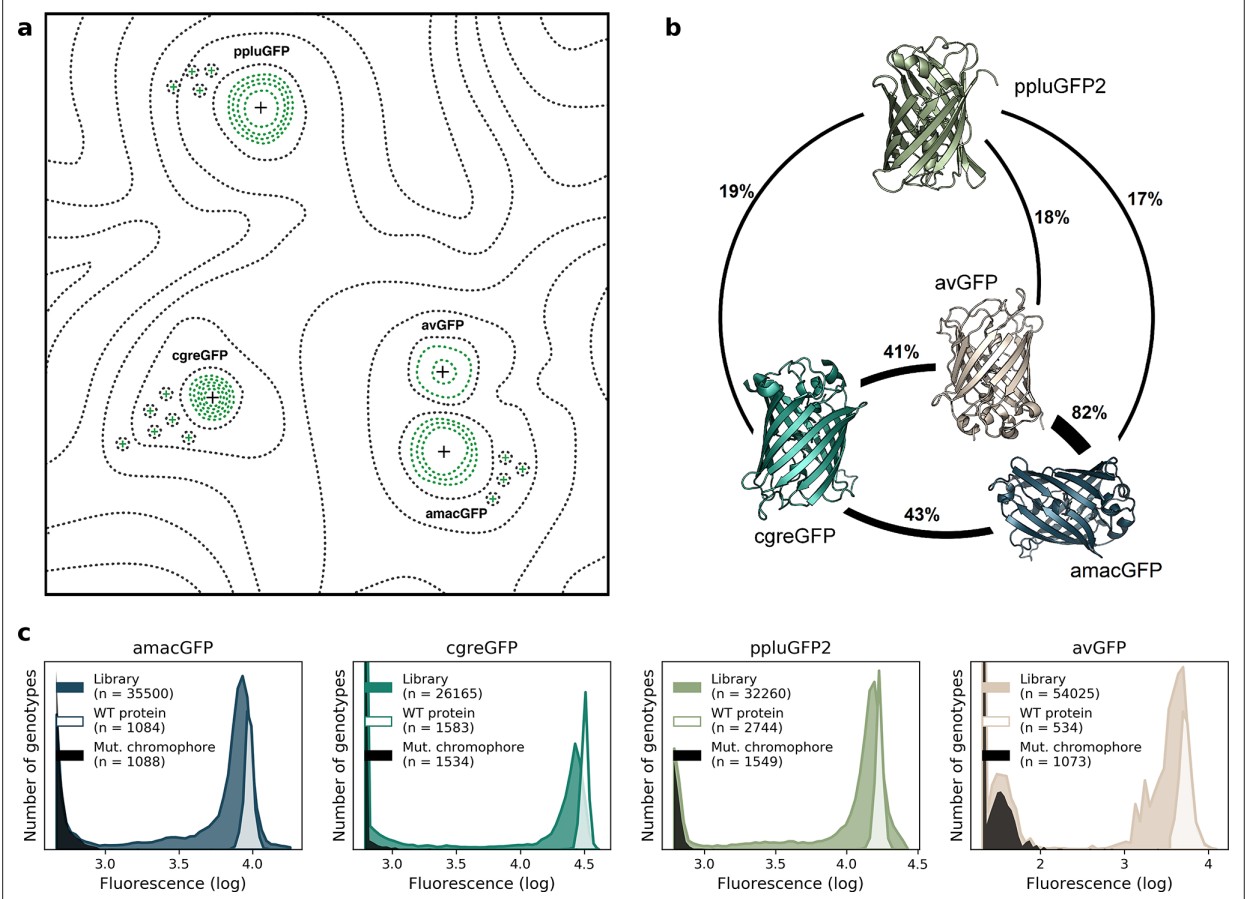

**Figure 1.** Comparison of four GFP fitness peaks. (**a**) A conceptual representation of the GFP fitness landscape following the visualization proposed by *Wright, 1932*. The black dotted lines represent the unknown regions of the fitness landscape and the green lines the surveyed local fitness peak. Wildtype GFPs (black +) and the predicted functional GFPs (green +) are shown at an approximate scale of sequence divergence from each other. Between the four wildtype fitness peaks there must also be some unknown but large number of other functional and wildtype GFP sequences. For clarity, we do not draw them in the figure. (**b**) Amino acid sequence identity between different orthologs, displayed in percent. (**c**) Distribution of fluorescence of mutant libraries (colour), control wildtype protein sequences (white), and protein sequences containing loss-of-function mutations in the chromophore (black).

The online version of this article includes the following figure supplement(s) for figure 1:

**Figure supplement 1.** Distributions of wild-type protein genotypes with and without synonymous mutations.

**Figure supplement 2.** Effects of mutations across the GFP sequences.

**Figure supplement 3.** Mutational bias in datasets generated from different mutagenesis strategies.

space has not been explored (*Chan et al., 2017*; *Kemble et al., 2019*). Furthermore, the exploration of the fitness landscape of specific proteins is one of the approaches in protein engineering (*Bryant et al., 2021*; *Romero and Arnold, 2009*; *Russ et al., 2020*; *Wittmann et al., 2021*). Such studies explore the fitness landscape of the protein of interest through deep mutational scan of a known protein sequence. This information is then used to predict novel functional protein sequences that are designed by introducing mutations into the original sequence. Here, we explored the interplay of the heterogeneity of fitness peaks of orthologous sequences and prediction of novel functional protein sequences (*Figure 1a*). To this end, we compared the fitness peaks of four GFPs that had different levels of sequence divergence from each other. We then used this information to accurately predict novel functional GFPs at considerable sequence divergence to any known GFP sequence.

**Table 1.** The dataset in numbers.

The avGFP data is from *Sarkisyan et al., 2016*.

| Gene | amacGFP | cgreGFP | ppluGFP2 | avGFP |
|---|---|---|---|---|
| Number of protein genotypes surveyed | 35,500 | 26,165 | 32,260 | 51,715 |
| Average (median) number of AA substitutions per genotype | 4.37 (3) | 4.23 (3) | 3.7 (2) | 3.93 (4) |
| Average (median) number of barcode replicates per protein genotype | 8.7 (5) | 6.8 (5) | 12 (7) | 1.2 (1) |
| Amino acid identity | avGFP: 82% cgreGFP: 43% ppluGFP2: 17% | avGFP: 41% amacGFP: 43% ppluGFP2: 19% | avGFP: 18% amacGFP: 17% cgreGFP: 19% | amacGFP: 82% cgreGFP: 41% ppluGFP2: 18% |
| False positive rate* | 0.55% (9 of 1635) | 0.75% (14 of 1860) | 0.49% (11 of 2242) | 0.24% (2 of 839) |
| False negative rate* | 0% (0 of 1084) | 0% (0 of 1583) | 0% (0 of 2744) | 0.08% (2 of 2444) |
| Mean wildtype log10 fluorescence level ± standard deviation | 3.97±0.031 (3.96±0.030 for amacGFP:V12L) | 4.50±0.028 | 4.23±0.027 | 3.72±0.082 |
| Fraction of genotypes in which epistasis cannot be ascertained† | 7.4% | 15.9% | 4.5% | 16.5% |
| Fraction of genotypes displaying \|epistasis\|>0.3 (>1) ‡ | 5.3% (0.2%) | 14.4% (5.6%) | 6.8% (0.9%) | 21.4% (11.6%) |
| Mutational LD50, loss of function § | 5.8 (5.7 for amacGFP:V12L) | 3.2 | 6.2 | 4.1 |
| Mutational LD50, loss of wildtype-level fluorescence level § | 1.7 (1.8 for amacGFP:V12L) | 0.9 | 1.7 | 2.2 |
| Proportion of machine-learning predicted genotypes displaying epistasis <–0.3 (<-1) | 78% (46%) | 57% (21%) | 81% (64%) | NA |

*False positive rates refer to the fraction of genotypes which are expected to be dark or dim due to chromophore mutations but which were assigned a bright fitness; false negative rates refer to genotypes encoding wildtype protein which were assigned dim or dark fitnesses.

†Calculation of epistasis requires knowledge of a genotype's expected fluorescence, i.e. the sum of contributions of individual mutations. For genotypes with multiple mutations, all individual mutations comprising the genotype must have been measured in isolation.

‡An absolute epistasis value of 0.3 or 1 implies a two-fold or ten-fold difference between the observed and expected fluorescence levels, respectively.

§"Mutational LD50, loss of function" refers to the number of mutations at which 50% of genotypes are rendered non-functional (i.e. assigned to the darkest FACS gate), obtained by fitting a logistic curve to the fraction of non-functional genotypes at each mutational step (see values in **Supplementary file 1**) and solving for f(x)=0.5; "Mutational LD50, loss of wildtype fluorescence level" refers instead to the number of mutations at which 50% of genotypes maintain a fluorescence level within two standard deviations of the WT level.

## Results

To complement the available data on the avGFP fitness peak (*Sarkisyan et al., 2016*; GFP from *Aequorea victoria*, Hydrozoa), we experimentally characterized three additional GFP sequences, each with a different degree of sequence divergence from avGFP: amacGFP (*Aequorea macrodactyla*, Hydrozoa), cgreGFP (*Clytia gregaria,* Hydrozoa), and ppluGPF2 (*Pontellina plumata*, Copepoda), with 18%, 59%, and 82% sequence divergence, respectively (*Figure 1b*; *Table 1*). For simplicity, we refer to all of these sequences as 'wildtype', even though only cgreGFP and ppluGPF2 were identical to the true wildtype sequences, while avGFP and amacGFP contain one and three amino acid substitutions, respectively. amacGFP, ppluGFP2, and cgreGFP were subject to a similar experimental pipeline (*Figure 2*) as avGFP (*Sarkisyan et al., 2016*). For each sequence a library of genotypes containing random mutations in the respective GFP sequence was generated by error-prone PCR, in which each GFP gene variant was labelled downstream of its stop codon by a primary barcode, a random combination of nucleotides. This mutant library was expressed as a fusion protein with the red fluorescent protein mKate2 in *E. coli* cells, which were then sorted based on green fluorescence intensity within a narrow red fluorescence gate, to control for gene expression level and other errors (*Figure 2—figure supplement 1*). The DNA barcodes of the sorted cells were sequenced and these data were used to perform a statistical analysis estimating the level of fluorescence of tens of thousands of GFP genotypes. Three notable improvements to the original experimental pipeline were implemented: gene sequence-agnostic library sequencing, genome integration of the construct, and use of secondary barcodes that introduced internal replicas in the experiment (see Materials and methods). These

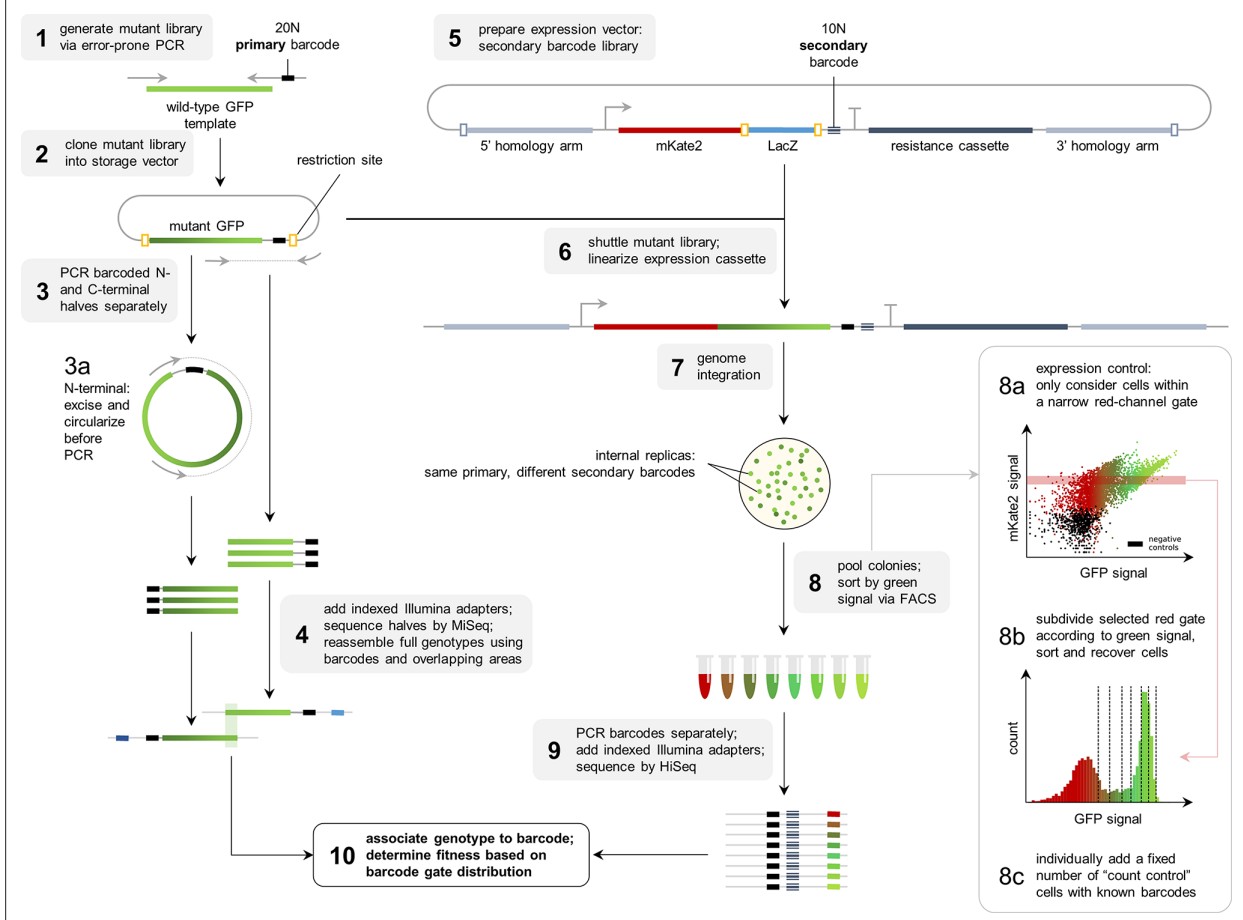

**Figure 2.** Flowthrough of the experimental methodology.

The online version of this article includes the following figure supplement(s) for figure 2:

**Figure supplement 1.** Distribution of cells during FACS sorting.

changes resulted in more physiologically relevant expression levels, made the pipeline more scalable, and reduced the variance of fluorescent genotype measurements by a factor of 7 (*Figure 1c*; *Table 1*). The new dataset contained 25,000–35,000 genotypes per each of the three additional fitness peaks, with each mutant genotype harboring on average 3–4 mutations relative to its respective wildtype sequence (*Table 1*). These data, together with data from avGFP, were then used in our comparative study of the GFP fitness peaks (*Figure 1a*).

The four fitness peaks shared substantial similarities (also see *Biswas et al., 2018* for sfGFP). In all cases, synonymous variants had no measurable effect on fitness, which may be a consequence of the experimental design aimed to be insensitive to expression levels and, thus, they were pooled for all subsequent analyses (*Figure 1—figure supplement 1*). Mutations in the chromophore eliminated fluorescence (*Figure 1—figure supplement 2*) and mutations of buried amino acid residues had a stronger effect than mutations of residues on the protein surface (*Figure 3b*; *Figure 1—figure supplement 2*). In all four fitness peaks, a threshold effect of accumulating multiple random mutations was found, such that the median level of fluorescence dropped sharply once a certain number of mutations was reached (*Figure 3a*; *Supplementary file 1*). The fitness peak shape differed substantially among different GFP sequences. Only 3–4 mutations were necessary for avGFP and cgreGFP, so the corresponding fitness peaks were sharp (*Table 1*). By contrast, the fitness peaks of amacGFP and ppluGFP2 were substantially flatter, with each tolerating twice as many mutations (*Figure 3a*). Furthermore, we compared the sharpness of the wildtype fitness peaks with the fitness peaks corresponding to sequences harbouring a single mutation relative to the wildtype sequence. Fitness peaks of most single-mutation neighbours with high levels of fluorescence were sharper than the respective wildtype

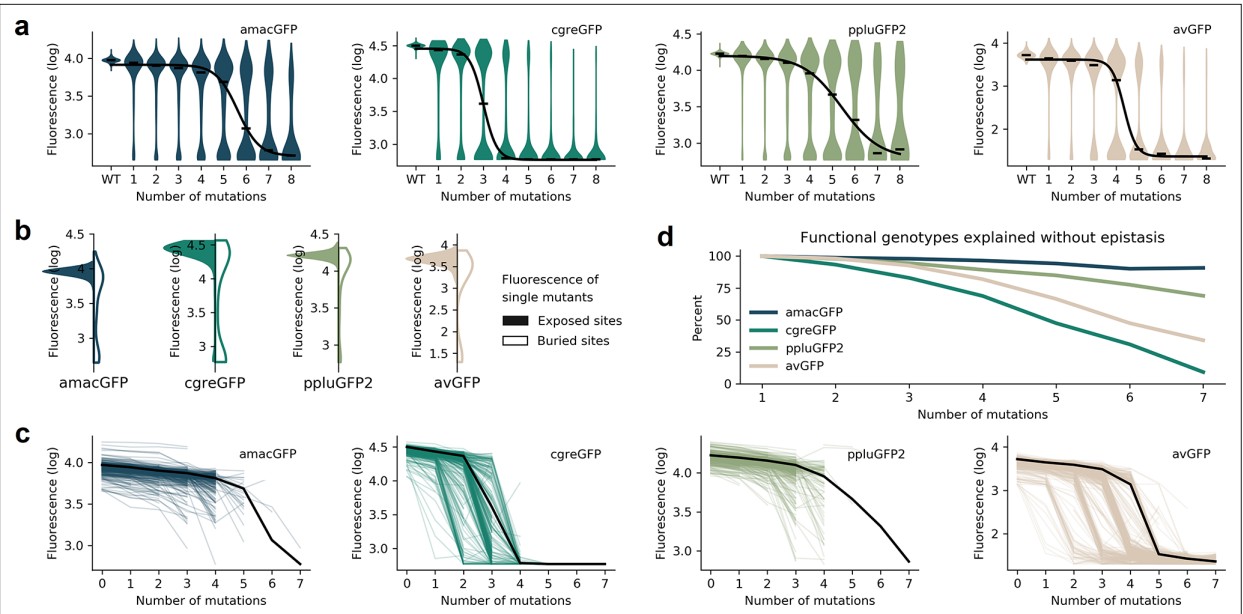

**Figure 3.** Distributions of fluorescence. (**a**) Fluorescence level distributions of genotypes at varying distances from the wildtype and the logistic curves fitted to the median fluorescence for each category (black line). (**b**), Distribution of fluorescence of genotypes with a single amino acid mutation at exposed (colour) versus buried (white) sites. (**c**) Starting from genotypes with one mutation away from the wildtype sequence, each line represents the median fluorescence as a function of sequence divergence away from such genotypes. Only points with at least 15 available genotypes are shown. The median fluorescence level at varying distances for the wildtype sequence, black lines, are shown for comparison. (**d**) The fraction of genotypes without epistasis was calculated as the ratio of the number of observed functional genotypes divided by and the number of genotypes expected to be functional under the assumption of no epistatic interactions between amino acid sites. In the absence of epistasis, the expectation is a constant value of 100% independent of the number of amino acid changes relative to the wildtype.

The online version of this article includes the following figure supplement(s) for figure 3:

**Figure supplement 1.** Epistatic interactions of mutations in GFP.

**Figure supplement 2.** Effect of extant and non-extant mutations.

fitness peaks (*Figure 3c*), suggesting a local optimization of robustness of each wildtype sequence (*Draghi et al., 2010*; *Zheng et al., 2020*; *Draghi et al., 2010*). Notably, the shape of the wildtype fitness peak showed no straightforward relationship with its respective level of fluorescence (*Table 1*), as may have been expected (*Johnson et al., 2019*).

We compared the fluorescence of each genotype to the expected level under an assumption that each mutation influences fluorescence level independently, that is without any epistasis (*Equation 1*):

$$epistasis \ = \ Effect_{observed} \ - \ Effect_{expected} \ = \ \left( F_m \ - \ F_{wt} \right) - \sum_i \left( F_i \ - \ F_{wt} \right) \cdot x_i \qquad (1)$$

where $F_i$, $F_m$ $F_{wt}$ are measured levels of fluorescence of a genotype with a single mutation $i$, of genotype $m$, or of the wildtype sequence, respectively, and $x_i = 1$ when mutation $i$ is contained within the genotype $m$ and $x_i = 0$ when it is not. We then calculated the fraction of genotypes that do not require epistatic interactions to predict their fluorescence. On all four fitness peaks, genotypes with two mutations away from the wildtype sequence rarely exhibited any epistatic interactions. However, a notable difference between the fitness peaks was observed when considering genotypes with multiple mutations. The level of fluorescence for a vast majority of genotypes with >5 mutations cannot be explained without epistasis in sharp fitness peaks, avGFP and cgreGFP. By contrast, few genotypes with >5 mutations in flat fitness peaks required epistasis to explain their fluorescence level, with amacGFP requiring almost no epistasis at all (*Figure 3d*, *Figure 3—figure supplement 1*). Interestingly, the sharpness of the fitness peaks and the concomitant extent of epistatic interactions did not correlate with the sequence divergence between the fitness peaks. Indeed, the two closest sequences (82% identity), derived from the same genus, are the sharp, epistatic avGFP peak and the flat, non-epistatic amacGFP peak (*Figure 3d*).

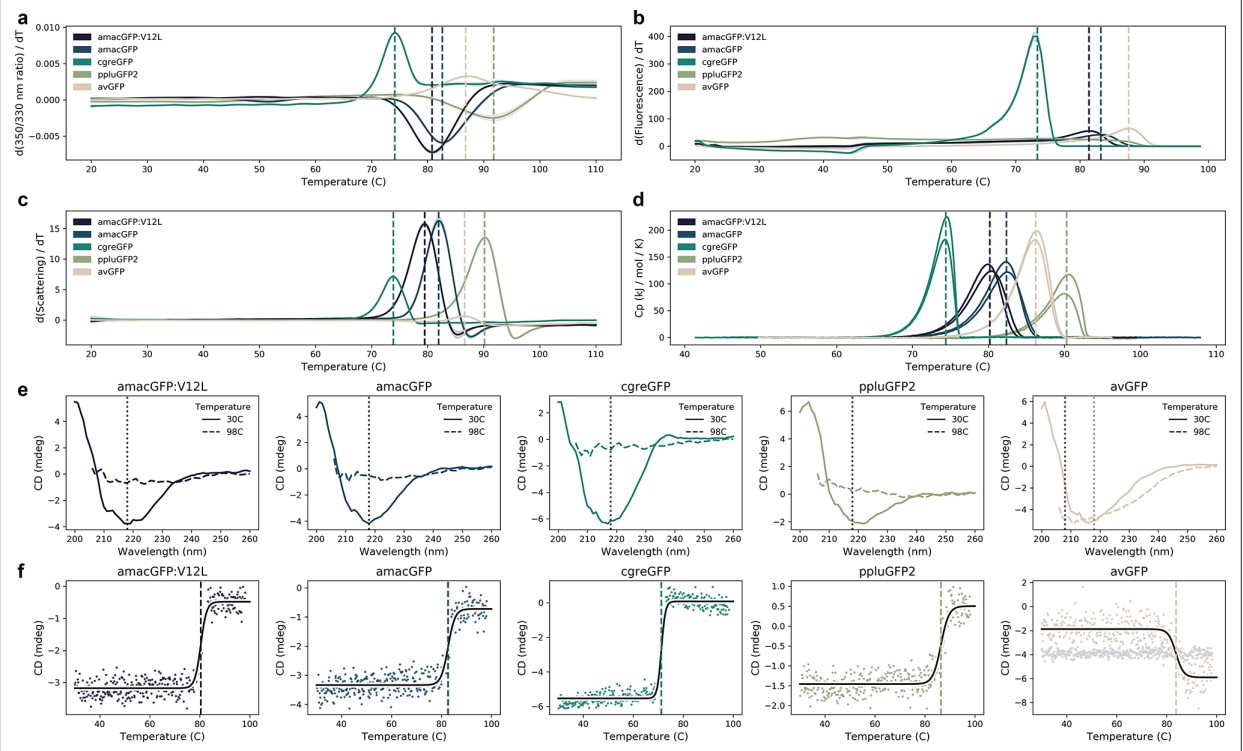

**Figure 4.** Thermal sensitivity of GFP orthologs. (**a**) Thermal unfolding measured by differential scanning fluorimetry (DSF) showing the first derivative of the ratio of 350/330 nm emission. Shaded areas indicate standard deviation of triplicates. (**b**) Melting curves of green fluorescence emission (510 nm) as a function of temperature measured on a qPCR machine. Shaded areas indicate standard deviations of eight technical replicates. (**c**) Thermal aggregation measured by DSF showing the first derivative of the light scattering. Shaded areas indicate standard deviation of triplicates. (**d**) Specific heat capacities measured by differential scanning calorimetry in duplicate. (**e**) Circular dichroism (CD) spectra measured before (30 °C) and after (98 °C) with the melting curves depicted in (**f**) where vertical dotted lines indicate the monitored wavelength in (**f**). f, CD melting curves monitored at 218 nm (and additionally 208 nm in the case of avGFP, where 218 nm did not show a transition), fitted with a logistic curve. In (**a**), (**b**), (**c**), (**d**), (**f**), vertical dashed lines indicate the melting temperature, except ppluGFP2 in (**b**). In (**a**), (**b**), (**d**), (**f**), temperature was increased at a rate of 1 °C per minute, in (**c**), at a rate of ~2 °C per minute, the slowest allowed by the LightCycler.

The online version of this article includes the following figure supplement(s) for figure 4:

**Figure supplement 1.** Urea denaturation and refolding of orthologues.

**Figure supplement 2.** Aggregation and oligomeric states in GFP orthologues.

**Figure supplement 3.** Correlation between fluorescence and ddG predicted by Rosetta.

**Figure supplement 4.** Effects of mutations in amacGFP and amacGFP:V12L.

**Figure supplement 5.** Spatial proximity of amino acid residues and detected pairwise epistasis.

Flat fitness peaks correspond to mutationally robust proteins, those that are capable of withstanding multiple mutations without losing function, while sharp fitness peaks correspond to mutationally fragile ones. The observed differences in mutational robustness of different proteins may be explained by thermodynamic stability (*Bershtein et al., 2006*; *Echave and Wilke, 2017*; *Gong et al., 2013*; *Kurahashi et al., 2018*; *Poelwijk et al., 2019*; *Sarkisyan et al., 2016*). Therefore, we performed an array of assays aimed at the biophysical characterisation of the four wildtype proteins and an additional genotype, amacGFP:V12L, which differed from amacGFP by the V12L mutation that was extremely common in the amacGFP mutant library. We have assayed the thermal stability of the proteins, using Differential Scanning Fluorimetry (DSF), Differential Scanning Calorimetry (DSC), Circular dichroism (CD), as well as simple measurements of fluorescence in a qPCR machine at different temperatures. We also assayed refolding kinetics of urea-denatured proteins (*Pédelacq et al., 2006*). Finally, we assessed oligomeric states of each of the proteins using multi-angle light scattering with size-exclusion chromatography (SEC-MALS).

**Table 2.** Biophysical and biochemical characterisation of wildtype GFPs.

| | amacGFP:V12L | amacGFP | cgreGFP | ppluGFP | avGFP |
|---|---|---|---|---|---|
| Unfolding Tm (DSF) | 80.8 °C | 82.6 °C | 74.1 °C | 91.8 °C | 86.8 °C |
| Aggregation Tm (DSF) | 79.5 °C | 82.0 °C | 73.9 °C | 90.2 °C | 86.6 °C |
| Tm (CD) | 80.4 °C | 82.6 °C | 71.2 °C | 86.4 °C | 83.7 °C |
| Transition slope (CD) | 0.86 | 0.72 | 1.27 | 0.63 | 0.67 |
| Tm (DSC) | 80.2 °C | 82.4 °C | 72.9 °C | 90.3 °C | 86.3 °C |
| Enthalpy of denaturation (DSC) | 744 kJ/mol | 768 kJ/mol | 755 kJ/mol | 515 kJ/mol | 1012 kJ/mol |
| Fluorescence loss Tm (qPCR) | 81.1 °C | 82.6 °C | 72.9 °C | - | 87.5 °C |
| Urea denaturation: initial rate* | −0.87 | −0.35 | −0.18 | −0.02 | −0.009 |
| Kinetic parameters for urea denaturation curves* | $a_1=0.71$ $k_1=0.96\ h^{-1}$ $a_2=0.28$ $k_2=0.25\ h^{-1}$ | $a_1=0.52$ $k_1=0.54\ h^{-1}$ $a_2=0.43$ $k_2=0.12\ h^{-1}$ | - | $a_1=0.92$ $k_1=0.02\ h^{-1}$ | $a_1=0.92$ $k_1=0.01\ h^{-1}$ |
| Refolding: initial rate† | 0.01 | 0.01 | 0.000014 | 0.05 | 0.007 |
| Kinetic parameters for refolding curves† | $a_1=-0.35$ $k_1=0.025\ s^{-1}$ $a_2=-0.36$ $k_2=0.005\ s^{-1}$ $a_3=-0.38$ $k_3=0.001\ s^{-1}$ | $a_1=-0.057$ $k_1=0.057\ s^{-1}$ $a_2=-0.39$ $k_2=0.013\ s^{-1}$ $a_3=-0.63$ $k_3=0.002\ s^{-1}$ | $a_1=0.16$ $k_1=0.036\ s^{-1}$ $a_2=-0.45$ $k_2=0.01\ s^{-1}$ $a_3=-0.87$ $k_3=0.001\ s^{-1}$ | $a_1=-0.32$ $k_1=0.14\ s^{-1}$ $a_2=-0.45$ $k_2=0.02\ s^{-1}$ $a_3=-0.21$ $k_3=0.003\ s^{-1}$ | $a_1=-0.4$ $k_1=0.016\ s^{-1}$ $a_2=-0.36$ $k_2=0.001\ s^{-1}$ $a_3=-0.31$ $k_3=0.001\ s^{-1}$ |
| Expected monomer size | 28.1 kDa | 28.1 kDa | 27.4 kDa | 25.7 kDa | 27.9 kDa |
| Primary oligomeric state (SEC-MALS) | Monomer (67%), dimer (31%) | Monomer (51%), dimer (46%) | Dimer (>99%) | Tetramer (>97%) | Monomer (>99%) |

*Curves monitoring loss of fluorescence in 9 M urea were fitted with two exponential functions in the case of amacGFP and amacGFP:V12L and one exponential function for avGFP and ppluGFP2, while cgreGFP fluorescence loss could not be well modeled using only exponential functions (see Figure 4—figure supplement 1). Initial rates were estimated by calculating the derivative at time t=0.

†Curves monitoring the recovery of fluorescence after urea denaturation over the course of 20 minutes were fitted with three exponential functions (see Figure 4—figure supplement 1). Initial rates were estimated by calculating the derivative at time t=0.

The different methods yielded complementary results (*Figure 4*; *Table 2*). Specifically, we observed that the most mutationally fragile protein, cgreGFP is also the most kinetically unstable protein and the most mutationally robust protein, ppluGFP2, was also the most kinetically stable (*Table 2*; *Figure 4*; *Figure 4—figure supplement 1*). These data tentatively suggest that the shape of the GFP fitness peaks, as characterized by mutational robustness, may in part be shaped by the underlying protein stability. This relationship does not appear to be perfect, as the mutationally fragile avGFP is stable, while amacGFP has mutational robustness comparable to ppluGFP2 (*Table 1*), but a substantially lower stability (*Table 2*). Indeed, there may be other factors that influence this relationship, such as the impact of protein folding on the GFP chromophore maturation, GFP folding histeresis (*Andrews et al., 2009*), the oligomeric protein state and the propensity of the mutant genotypes to aggregate. Indeed, avGFP is the only exclusive monomer from among the four wildtype sequences (*Table 2*; *Figure 4—figure supplement 2*) while ppluGFP2 is exclusively tetrameric. The propensity for aggregation also appears variable between the genotypes, with amacGFP showing the highest aggregation of non-fluorescent genotypes (*Figure 4—figure supplement 2*). Furthermore, our measurements of GFP thermostability through refolding may not reflect GFP folding as it occurs in vivo in the course of translation.

We then used a computational approach to further explore the relationship between protein stability and the shape of the fitness landscapes. We solved the crystal structure of amacGFP and analysed it along with structures already available for other proteins. We found that mutations causing a substantial reduction of fluorescence tended to have a higher effect on protein stability (*Figure 4—figure supplement 3*), estimated by predicted ΔΔG (Two-sided Mann Whitney U test, $p<10^{-6}$). Furthermore, we found a statistically significant correlation between predicted ΔΔG and the effect of a mutation, which was stronger in sharp fitness peaks, avGFP and cgreGFP, and weaker in the flat fitness peaks, amacGFP and ppluGFP2 (*Figure 4—figure supplement 3*; Spearman's correlation $r=0.6$ and $r=0.3$, respectively). Interestingly, the V12L mutation in amacGFP:V12L appears to have shifted the

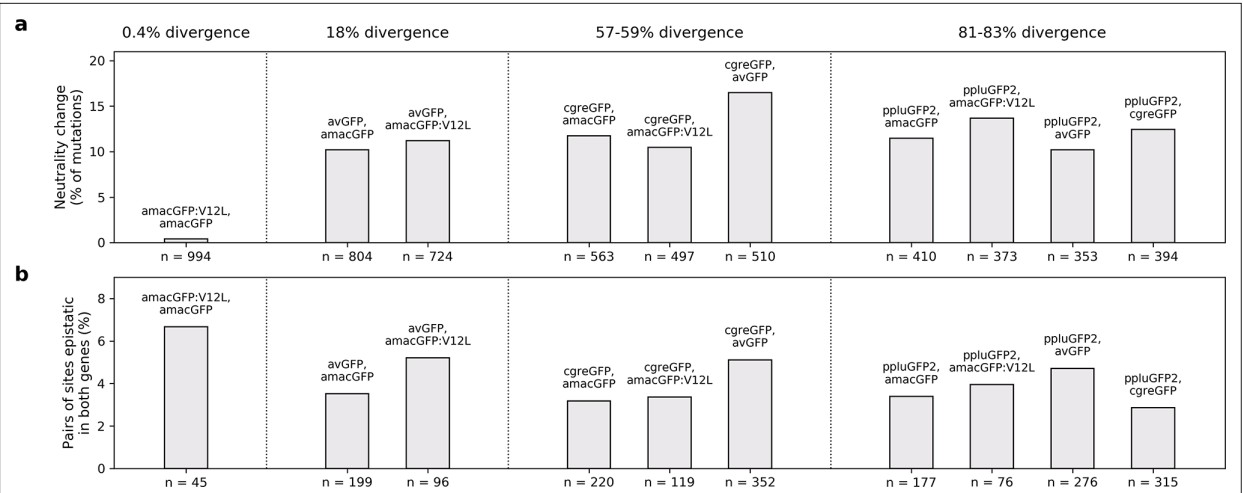

**Figure 5.** Differences in mutational effects in GFP orthologues. (**a**) The proportion of single amino acid mutations which were observed to be neutral (maintaining fluorescence within two standard deviations of the wildtype level) in one GFP sequence and deleterious (reducing fluorescence by over five standard deviations) in another GFP, out of all mutations surveyed in both. The total number of amino acid states considered is indicated beneath the bars. (**b**), For each pairwise GFP comparisons, first we selected all pairs of amino acid sites for which epistasis was measured. Then, out of these pairs of sites we calculated the number that had epistasis >0.3 in either of the two GFP genes (reported underneath each bar). Finally, we calculated the percent of epistatically interacting sites that were measured to be epistatically interacting in both (y-axis). In (**a**) and (**b**), pairs of genes are arranged in order of increasing sequence divergence.

distribution of the mutation effects, substantially increasing the effect of mutations on the barrel lid in proximity to residue 12, without impacting the overall mutational robustness (**Figure 4—figure supplement 4**). Across the whole landscape, epistatically interacting amino acid residues were slightly more likely to be spatially proximal (**Melamed et al., 2013**; **Sarkisyan et al., 2016**) and the effect was more pronounced in the flatter fitness peaks (**Figure 4—figure supplement 5**). Taken together, these data suggest that the heterogeneity in the shape of the orthologous GFP fitness peaks may be related to the stability of the underlying protein sequences.

The apparent lack of a relationship between sequence divergence and fitness peak shape suggests that the shape changes on a scale that is smaller than the distances between the four GFP proteins. Therefore, the difference of the impact of mutations on different fitness peaks should be independent from the sequence divergence between them. We found that the probability that a neutral mutation in one protein becomes deleterious in another one was independent of the sequence divergence, except when considering two protein sequences that were different by a single amino acid change (**Figure 5a**). A complementary pattern was observed when we considered if there is a difference in which pairs of sites are interacting epistatically. The probability that a pair of epistatically interacting sites in one protein is also epistatically interacting in another protein was largely independent of sequence divergence between the two proteins, with the two highly similar sequences showing the highest degree of similarity (**Figure 5b**). Taken together, these data indicate that underlying rules that determine epistatic interactions and fitness peak shape change on a scale smaller than 20% of sequence divergence.

The identification of two mutationally robust proteins presented an opportunity to predict novel GFP sequences. Two lines of reasoning led us to hypothesize that it would be easier to create functional genotypes by introducing mutations into mutationally robust, rather than fragile, proteins. First, robust proteins had a higher fraction of fit genotypes with >5 mutations and, therefore, it should be easier to find other genotypes that are farther away. Second, a robust protein should be more tolerant of mistakes in predictions.

Prediction of functional genotypes many mutations away from known functional sequence is akin to looking for a needle in a haystack. There are $\binom{222}{48} \cdot 19^{48}$ or $\sim 10^{110}$ genotypes that are 48 mutations away from a 222 amino acid long ppluGFP2. Out of all of these sequences, only an infinitesimally small proportion is expected to be functional, perhaps as few as $10^{-11}$ (**Keefe and Szostak, 2001**) and finding any appreciable number of these sequences requires extraordinary precision. Therefore,

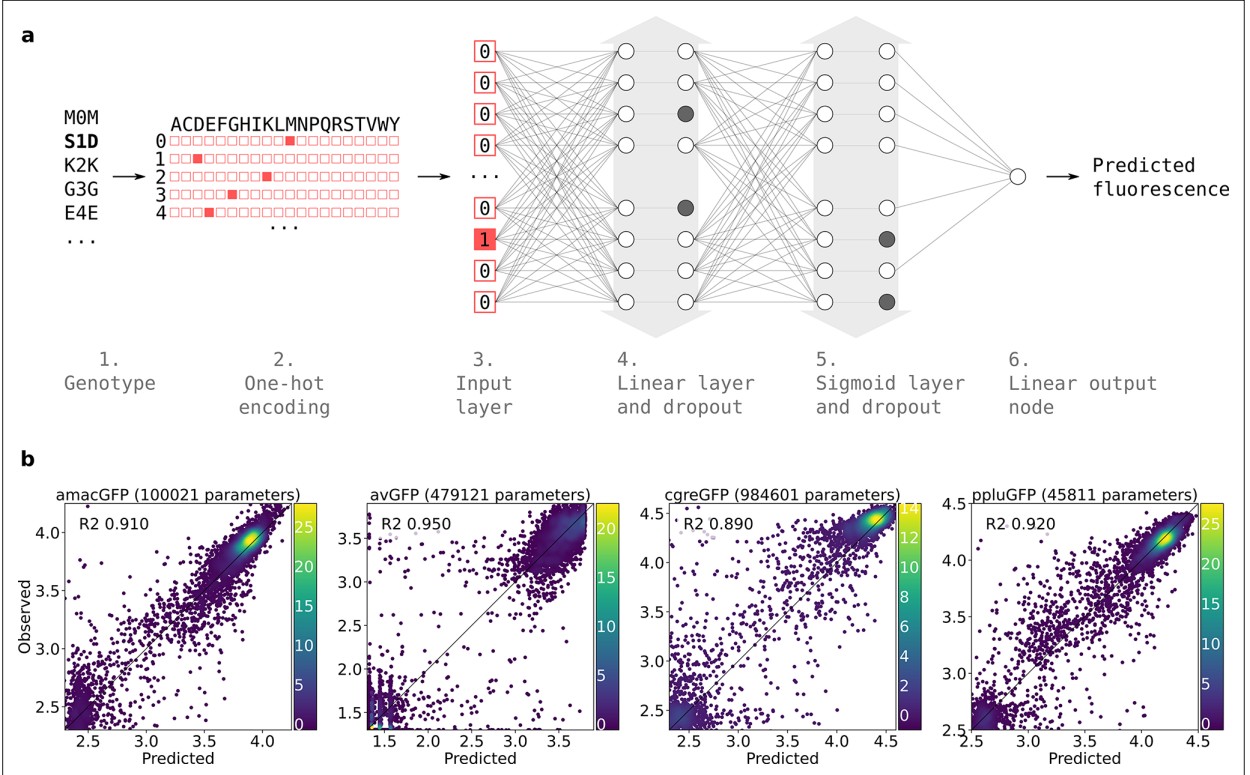

**Figure 6.** Neural network structure. (**a**) 1. Each genotype in the dataset was denoted by the mutations it contained relative to its parental wildtype sequence. 2. Genotypes were one-hot encoded. For each position in the sequence, a binary vector indicated present (red = 1) and absent (white = 0) amino acid states. 3. One-hot encoded sequences were flattened and provided to the neural network as input. 4. The first hidden layer contained linear nodes followed by a dropout layer of the same size. 5. The second hidden layer contained sigmoid nodes followed by a dropout layer of the same size. Grey arrows indicate layer widths that were optimised by a random search. Greyed-out neurons without output connections represent randomly inactivated neurons in dropout layers. During training, randomly inactivated neurons prevented overfitting. At inference time, randomly inactivated neurons allowed the model to provide different estimates of the fluorescence each time a prediction was run on a genotype. 6. Linear node outputting predicted fluorescence values. For each predicted genotype, the median of several fluorescence estimates was used as the final fluorescence level. (**b**) Correlations between observed and predicted levels of fluorescence with an optimized architecture. Datapoint density is represented in color.

The online version of this article includes the following figure supplement(s) for figure 6:

**Figure supplement 1.** Correlations between observed and predicted levels of fluorescence.

we used a machine learning approach, training neural networks on the genotype-to-phenotype relationships revealed by our data (see Materials and methods, *Figure 6*). We split this data into non-overlapping training and validation sets. Models were trained on the training set and after training, model goodness was calculated as the coefficient of determination between predicted and actual fluorescence values for all genotypes in the validation set. We started with a linear model fitted to the one-hot encoded protein sequences. The validation score of the resulting models indicated that between 59% and 82% of the variance could be explained in all landscapes by the simple linear contribution of mutations in the protein sequence (*Figure 6—figure supplement 1*). This simple estimate of the fluorescence, which is called fitness potential (*Kimura and Crow, 1978*; *Milkman, 1978*), is simply the summed contribution of weighted mutations and does not account for possible interactions between them. We then trained models of increasing capacity and aimed at maximising the validation score while reducing overfitting. In all landscapes, the majority of genotypes were either non-functional or of near-wildtype brightness; this scarcity of genotypes with intermediate fluorescence levels suggests that an abrupt threshold function transforms the fitness potential into the final fluorescence level, as has been observed previously (*Pokusaeva et al., 2019*; *Sarkisyan et al., 2016*). Therefore, we decided to train sigmoid models, resulting in the successful capture of an additional 13%, 78%, 53%, and 39% of the remaining unexplained variance for amacGFP, avGFP, cgreGFP, and ppluGFP2, respectively, compared to the results of the linear model (*Figure 6—figure supplement*

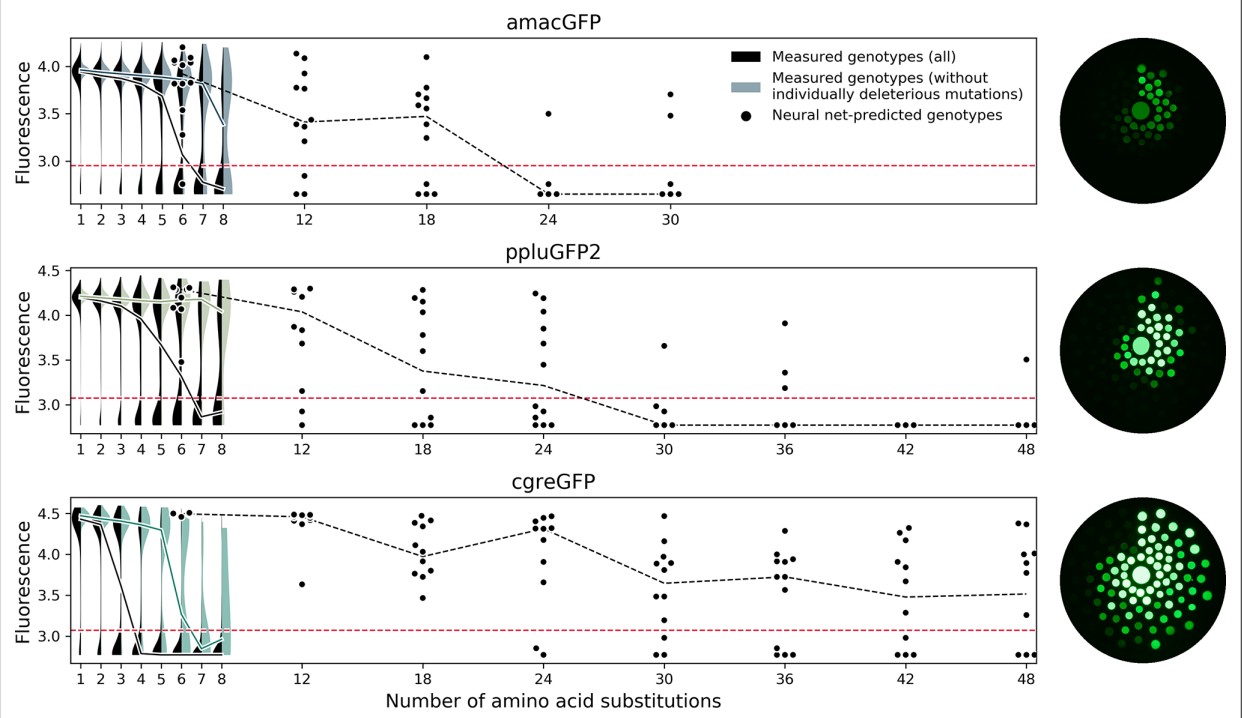

**Figure 7.** Predicting functional GFP mutants. Violin plots show the distribution of fluorescence of all genotypes (black) and combinations of only individually neutral mutations (color). Experimental measurements of the level of fluorescence in genotypes predicted by the neural network are shown as black dots (12 genotypes per distance). Black dashed lines show the median fluorescent values for each group. Red dashed lines indicate the cutoff of detectable fluorescence. Photos of agar plates with *E. coli* spots expressing predicted GFP variants are shown on the right. Spots of bacteria expressing GFP variants are arranged in circles around the wildtype gene at increasing distance with the number of mutations (6, 12, 18, 24, 30, 36, 42, 48 mutations). For each group of genotypes, the brightest ones were inoculated at the top, with fluorescence decreasing clockwise.

The online version of this article includes the following figure supplement(s) for figure 7:

**Figure supplement 1.** Mutations used in machine learning-generated genotypes.

1). This minute transformation of the fitness potential noticeably improves the models' power, especially for the two genes that display the highest levels of epistatic interactions, avGFP and cgreGFP. To capture the functions that transform the fitness potential into the predicted fluorescence, we decided to train models with an output subnetwork of several sigmoid nodes (***Figure 6—figure supplement 1***). These functions are shown in ***Figure 6—figure supplement 1***. Theorising that models accounting for interactions between residues would push further the predictive power of the models, we optimised the architecture of two-layered networks, one for each dataset using a grid search approach. This resulted in models capturing 0.88, 0.95, 0.86, and 0.90 of variance for amacGFP, avGFP, cgreGFP, and ppluGFP2 respectively, as shown in ***Figure 6b***. Using the trained model as the evaluation function of a genetic algorithm, we made fitness peak-specific predictions, using the data of each fitness peak to predict fluorescent genotypes containing up to 48 mutations relative to the wildtype sequence.

Amino acids observed in homologous sequences, or extant states, are more likely to be neutral when introduced into a sequence of interest (***Pokusaeva et al., 2019***; ***Figure 3—figure supplement 2***). Therefore, one approach to predict a novel functional sequence would be to prefer the introduction of extant amino acid states. However, we wanted to push the envelope of our predictions in exploring uncharted regions of the GFP fitness landscape, avoiding the genotype space between known GFP sequences (space between fitness peaks in ***Figure 1a***). Thus, we aimed to predict genotypes as distant as possible from any known functional GFP sequences, corresponding to an area of GFP genotype space not known to be explored by evolution. Therefore, for experimental verification from among the predictions made by the machine learning algorithm, we selected sequences with the maximum amino acid states that were observed in our libraries but not present in any natural GFP (***Source data 8***).

Contrary to our expectation, experimental verification showed that the accuracy of our predictions was substantially higher for genotypes predicted by using data from the sharp cgreGFP fitness peak (*Figure 7*). For genotypes with 48 mutations (>20% sequence divergence of GFP), our predictions had an 8% accuracy when using data for the mutationally robust ppluGFP2 and a 50–60% accuracy for the mutationally fragile cgreGFP (*Figure 7*). These results may be relatively trivial, if the predictions were based on universally neutral mutations (*Kondrashov and Kondrashov, 2015*), those that are neutral in any GFP sequence. However, three lines of evidence show that our high rate of prediction cannot be explained by universally neutral mutations (also see *Poelwijk et al., 2019*). First, individual mutations which were strongly deleterious in some genetic contexts were successfully incorporated into functional predictions (*Figure 7—figure supplement 1*), and these conditionally deleterious mutations comprised a significant fraction of all mutations used in predictions, particularly in the case of the epistatic cgreGFP protein (*Figure 7—figure supplement 1*). Second, the mutations used in successful predictions occur in evolution at a rate two times slower than neutral synonymous substitutions (0.057 dn rate vs 0.11 ds rate, respectively, two-sided Mann-Whitney U-test p<0.00001), demonstrating that they are under negative selection. Finally, successful identification of universally neutral mutations would lead to a successful prediction of distant derivatives of any GFP sequence, not just the mutationally fragile cgreGFP. Furthermore, the ML-designed variants derived from the more robust amacGFP and ppluGFP2 proteins were rendered non-fluorescent by negative epistasis substantially more frequently than those derived from the fragile and epistatic cgreGFP (*Table 1*). This indicates that the success of the neural network was dependent on being able to learn epistatic interactions from the data, which were abundant in cgreGFP but rare in amacGFP and ppluGFP2, and to avoid non-favorable epistatic interactions, rather than relying on universally neutral mutations.

## Discussion

Experimental survey of the fitness landscape of a protein of interest is increasingly used in protein engineering to discover novel sequences with specific functions (*Bryant et al., 2021*; *Romero and Arnold, 2009*; *Russ et al., 2020*; *Wittmann et al., 2021*). While this approach remains challenging for proteins with a function that cannot be easily ascertained in a high-throughput manner (*Romero and Arnold, 2009*), it is likely to be more widely used in the future due to technological advances of experimental (*Romero and Arnold, 2009*) and analytical (*Wittmann et al., 2021*; *Wu et al., 2019*) tools. Our description of heterogeneity of fitness peaks of orthologous GFPs suggests some practical considerations for such surveys of other proteins. Researchers applying such methods to their protein of interest will inevitably have to choose a specific protein sequence to experimentally assay (*Romero and Arnold, 2009*). When the goal is to discover as many distant functional proteins as possible (i.e. *Bryant et al., 2021*; *Romero and Arnold, 2009*; *Russ et al., 2020*; *Wittmann et al., 2021*) it may seem natural to select a structurally or mutationally robust protein. Indeed, a robust protein, one that is known to be able to maintain function upon the introduction of many mutations, seems a good starting point to introduce even more mutations. Our results counter this intuition, and our recommendation is to select a fragile protein as the original template for a mutational scan. For the data of the fitness landscape to be useful for a downstream model to predict distant sequences, it has to contain information about epistatic interactions between mutations. Thus, a useful fitness landscape should contain many genotypes that have been rendered non-functional through negative epistatic interactions among a handful of mutations. Our results for cgreGFP demonstrate this principle. Out of 188 genotypes six mutations away from cgreGFP that were expected to be functional by an additive model only 61 (32%) were actually fluorescent. The rest were non-fluorescent, revealing extensive negative epistasis among those mutations. By contrast, our model correctly predicted 100% (12/12) of genotypes 6 mutations away from cgreGFP, learning to avoid combinations of individually neutral mutations that combine to create a non-functional genotype. Without this information, such as for amacGFP that shows almost no epistatic interactions within the surveyed genotypes, the model cannot learn which genotypes to avoid (*Figure 7*). The reported ~20% prediction accuracy at 40 mutations for sfGFP is also consistent with the sharpness of its fitness peak (*Biswas et al., 2018*).

Without direct knowledge of mutational robustness of a protein sequence our data indicate that researchers may rely on thermodynamic stability to choose the initial template protein, although the relationship between mutational and thermodynamic robustness may be more complex (*Table 2*). However, given that for many proteins it is likely to be easier to measure stability than mutational

robustness, choosing a structurally unstable protein from several available candidates may prove to be an acceptable compromise.

Despite the high accuracy of prediction with our model, there are still substantial limits to the prediction of functional proteins. Indeed, the relatively accurate prediction of functional GFP sequences up to 20% divergence from cgreGFP does not imply an ability to predict all $\binom{235}{48} \cdot 19^{48}$ or ~$10^{112}$ possible functional sequences at this level of divergence. The substantial heterogeneity between fitness peaks of the highly similar avGFP and amacGFP (18% divergence) suggests that predictions based on a single fitness peak may have lower accuracy of prediction of sequences not governed by the same set of epistatic interactions (*Alley et al., 2019*; *Lee et al., 2018*). However, the understanding of the heterogeneity of such predictions would require random sampling of all $10^{110}$ sequences, which is not presently feasible.

The four proteins in our study function in different species so the heterogeneity of their fitness landscape may be related to some aspect of the environment in which these species are typically found, such as temperature. By contrast, our experimental setup measures the fluorescence of these proteins in an identical, controlled but an artificial environment. If GFPs mutational robustness, fitness peak shape, was adapted to environmental conditions it implies that mutational robustness is driven by natural selection. Indeed, experimental data suggests that mutational robustness of GFP is a selectable trait (*Zheng et al., 2020*), so a correlation between sequence divergence and mutational robustness may have been expected. However, selectable traits are generally expected to evolve slowly, which translates to the expectation that similar sequences confer similar phenotypes. The lack of a correlation in our data between sequence similarity and mutational robustness implies that selection for the phenotype of mutational robustness in GFP is weak or that it changes rapidly on an evolutionary timescale.

The heterogeneity of the shape of the fitness peaks is remarkable. Up to 17% of all genotypes six random mutations away from the ppluGFP2 wildtype sequence have the same level of fluorescence as the wildtype. By contrast, only 0.9% of such genotypes derived from the fragile cgreGFP exhibited wildtype fluorescence (*Supplementary file 1*). However, it remains unclear whether this heterogeneity influences protein evolution. It is tempting to suggest that these data indicate that ppluGFP2 is a more 'evolvable' protein compared to cgreGFP. However, 2% of all genotypes 6 mutations away from wildtype cgreGFP were observed to be functional, which is still $2.10^{17}$ ($\binom{235}{6} \cdot 19^6 \cdot 0.02$) total functional genotypes, so even such a relatively fragile protein may not be restricted in its long-term evolution (*Povolotskaya and Kondrashov, 2010*). What fraction of all genotypes ~250 amino acids in length are functional GFPs, and what factors govern differences in the shape of fitness peaks of orthologous proteins, remain unknown.

## Materials and methods
### Selection of genes

Candidate fluorescent proteins were selected based on several criteria: fluorescence in the green spectrum, ability to mature and fluoresce in *E. coli* under standard culture conditions, and varying degrees of sequence divergence from each other. We also preferred candidates with an available solved crystal structure. Eight genes from six species (*Aldersladia magnificus*, *Aequorea macrodactyla*, *Clytia gregaria*, *Clytia hemisphaerica*, *Pontellina plumata*, and *Asymmetron lucayanum*) were selected as initial candidates. The *A. macrodactyla* protein contained three point mutations, as the wildtype was previously reported to mature poorly in *E. coli* (*Luo et al., 2006*). After testing their expression in *E. coli*, the three brightest proteins were chosen for further experiments: amacGFP, cgreGFP, and ppluGFP2, respectively from *A. macrodactyla*, *C. gregaria*, and *P. plumata*. Protein sequences of the chosen genes were aligned using the T-Coffee Expresso structural alignment (*Armougom et al., 2006*).

### Golden gate cloning

Single-step digestion/ligation Golden Gate protocols were adapted from *Weber et al., 2011*. All MoClo reaction mixtures contained 50 ng each of insert DNA and destination vector, 10 U of T4 DNA ligase (ThermoFisher), 20 U of type IIS restriction enzyme (BsaI, BpiI, or BsmBI; ThermoFisher), in T4 ligase buffer at a final concentration of 1 X and volume of 20 ul. Thermocycler conditions were as

follows: 10 min at 37°C, 25 cycles of 1.5 min at 16°C and 3 min at 37°C, 5 min at 50°C, and 10 min at 80°C.

## Generation of mutant libraries

Selected genes were ordered as synthetic dsDNA (Twist Biosciences), codon-optimized for bacteria and compatible with common modular cloning (MoClo) standards (*Weber et al., 2011*). For positions occupied by the same amino acid in different genes, the same codon was used in all genes. The same constant, 20-nucleotide region was included in each gene after the stop codon, for future primer-annealing purposes. All genes were cloned into non-expression storage vectors via MoClo. We generated mutant libraries of each gene via random mutagenesis with the Mutazyme II kit (Agilent), using 200 ng of DNA template and eight cycles of mutagenic PCR, in order to achieve an average of ~4 mutations per clone. Primers (Sigma) included type IIS restriction sites for later cloning, and 20 N random nucleotide barcodes to label each molecule with a unique identifier, hereafter referred to as 'primary barcode'. PCR product was gel-purified and cloned into a spectinomycin-resistant storage vector (MoClo Toolkit) via Golden Gate cloning, and transformed into high-efficiency chemically competent *E. coli* cells (Lucigen E. cloni 10 G); post-heat shock recovery time was limited to 15–20 min to avoid cell division during recovery and ensure that each resulting colony was the result of an independent transformation event. Up to 150 thousand colonies were recovered and pooled; DNA was extracted following standard maxiprep plasmid extraction protocol (ThermoFisher, GeneJet maxiprep kit) using 2–4 g of pooled colonies instead of liquid culture. Mutation rates were confirmed by Sanger sequencing (Microsynth) for random 25 clones per library prior to colony pooling.

The mutagenesis kit for creating mutant libraries for amacGFP, cgreGFP and ppluGFP2 was different than that used to create the avGFP mutant library, which led to a slightly different mutational signature, which has not affected our results (*Figure 1—figure supplement 3*).

## Generation of expression cassettes

We assembled an expression vector via MoClo from the following parts: a 5600 bp homology arm to the *E. coli* genome; an mKate2-LacZ fusion under the T5 promoter, followed by a placeholder sequence flanked by IIS restriction sites, and lambda T0 terminator; a zeocin resistance cassette; and a 3600 bp homology arm. The placeholder was subsequently replaced by 10 N random nucleotide barcodes, hereafter referred to as 'secondary barcodes', in order to create a library of around 10 thousand expression vector variants differing only by this barcode.

Mutant libraries were then shuttled from their storage vectors into the pooled expression vectors, replacing LacZ with GFP in-frame with mKate2 and allowing for color-based determination of cloning efficiency upon plating with X-Gal. Final constructs thus expressed GFP mutants as a fusion protein with mKate2, ensuring the two proteins are equimolar inside each cell and allowing mKate2's red fluorescence to be used as a control for GFP expression level (*Figure 2—figure supplement 1*). mKate2 was selected as previously (*Sarkisyan et al., 2016*) due to its spectral properties (minimum overlap with GFP spectra, and lack of green emission phase during maturation) and monomeric activity, and is separated from GFP by a rigid alpha-helix linker to avoid any potential interactions between the two proteins. An N-terminal 6-His-tag was also included in the mKate2-GFP fusion design.

GFP mutant libraries were shuttled into the expression vector via modular cloning protocols described above. MoClo reactions were transformed into high-efficiency chemically competent *E. coli* cells (Lucigen E. cloni 10 G). Around 800 thousand colonies were recovered; each mutant genotype, identifiable by its primary barcode, is thus expected to be associated with multiple secondary barcodes. This approach created internal replicates for each genotype, with each primary/secondary barcode combination having been the result of independent cloning and transformation events, allowing for independent measurements of the same genotype during a single experimental set-up.

## Genome integration

Genome integration is expected to produce less expression noise compared with expression from a plasmid (*Lee et al., 2015*). Final expression-ready cassettes were excised from the vector backbone via digestion at SpeI sites flanking the homology arms, and gel-purified. Linear fragments were integrated into a safe harbor in the *E. coli* chromosome via CRISPR-Cas9-mediated homologous recombination, using a protocol adapted from *Bassalo et al., 2016*. In brief, we transformed cells with Court

lab's pSIM5, a temperature-inducible plasmid containing genes necessary for homologous recombination, and pX2-Cas9 (Addgene #85811), an arabinose-inducible Cas9 vector. Cells were grown to the exponential phase ($OD_{600}$=0.6, measured via NanoDrop) at 30°C in the presence of 0.2% arabinose and then heat-shocked at 42°C for fifteen minutes to activate pSIM5. We observed increased efficiency when cells were grown with arabinose from the start, rather than only provided with it during the recovery phase. Cells were then placed on ice for 20 min, washed thrice with ice-cold distilled water, and electroporated with the linear library DNA as well as the SS9_RNA (Addgene #71656) vector containing the guide RNA for Cas9 to target the safe harbor. Cells were plated on 50 mg/L zeocin plates after two hours of recovery at 30°C in 0.2% arabinose-supplemented LB, grown overnight at 30°C and an additional day at room temperature, then recovered from plates and resuspended in LB for sorting. Approximately, 5 million colonies were recovered in each case.

## Fluorescence-activated cell sorting

Resuspended cells were sorted in parallel on two independent BD FACS Aria III cell sorters ('machine A' and 'machine B') at a rate of around 20 thousand events per second. Each library was processed independently, but a small amount of wild-type avGFP, amacGFP, cgreGFP, and ppluGFP2 genotypes with known barcodes were added to each library as positive controls and for the purposes of cross-library comparisons. A narrow gate in the red channel was selected, corresponding to a fixed mKate2 expression level, and this was subdivided into eight sub-gates based on green intensity (*Figure 2—figure supplement 1*) and these were sorted into eight separate tubes. For each library, around 28 million cells were sorted in total, leading to an estimated average of ~35 recovered cells for each colony in the pool.

After library sorting, we separately added 5000 cells each of four known barcodes to each tube, to serve as controls for the number of cells sorted and determine how many reads are generated per cell. 'Count control' cells were generated separately from the mutant libraries, so their barcodes are not expected to be present in any of the libraries.

The use of mKate2 controls for the influence of variability in gene expression, but also can be used to control for the impact of mutations on mRNA structure, stability, or translation. Such mutations could be either synonymous or non-synonymous. If mutations had a substantial probability of impacting green fluorescence through mRNA structure, stability or through their impact on translation, we would expect for our data to contain a non-negligible amount of such synonymous mutations. However, since synonymous mutations do not influence fluorescence (*Figure 1—figure supplement 1*), these effects are not present on a detectable level.

## Circularization of mutant libraries

For each gene, approximately 5 µg of DNA (mutant libraries in storage vectors) was digested with BsaI and the GFP fragment was recovered via gel purification, leaving known 5' overhangs. A short dsDNA filler sequence with compatible overhangs was used to tie the N- and C-terminal GFP ends together. The filler was obtained by annealing two complementary primers (mixed together in equal amounts, heated to 95°C, and gradually cooled down to 20°C at a rate of 0.1°C per second using a thermocycler); compatible overhangs with the GFP fragment were generated by BsaI digestion.

Circularization was performed at room temperature in a volume of 500 µl 1X T4 DNA ligase buffer, with 100 U BsaI and 60U T4 DNA ligase, and starting quantities of 50 ng each of linear GFP library fragment and dsDNA filler. Another 50 ng of each were added every 30 min until reaching a combined total amount of 2 µg. DNA addition was performed gradually in order to minimize the concentration of unligated linear DNA and thereby avoid formation of tandem multimers; once ligated, circular products cannot be cut again due to restriction site destruction. The circular monomer fraction was isolated by gel purification of the appropriate band. Successful circularization was confirmed by PCR.

## Preparation of mutant libraries for sequencing of coding region

Mutant GFP libraries were sequenced via MiSeq 300bp-paired-end Illumina sequencing, performed by the Vienna Biocenter Core Facilities. In order not to exceed the maximum total read length of 600 bp, N-terminal and C-terminal halves – each between 400 and 500 bp – were prepared separately.

C-terminal halves were PCR'd directly from the storage vectors containing the mutant libraries. For N-terminal halves, libraries were first circularized in order to place the barcode adjacent to the

start codon. In each case, a first round of 10 PCR cycles was performed, using three pairs of primers incorporating part of the constant region of Illumina TrueSeq adapters; the three pairs differed only by the addition of 1–2 N bases, in order to create sequence shifts and increase complexity for NGS purposes. These products were gel-purified and used as templates for a further 10 PCR cycles with primers incorporating Illumina indices. Final PCR products were gel-purified, eluted in nuclease-free water and sent for sequencing. Different indices were used for different halves and for different genes, allowing pooling of samples to be sequenced in the same MiSeq lane. A total of four lanes were used for MiSeq library sequencing.

## MiSeq data processing

Sample de-multiplexing was performed by the sequencing facility. Raw Illumina sequencing data was converted from .bam to .fastq format using Bamtools; all further processing was performed with custom Python scripts. BioPython was used for pairwise alignments.

MiSeq reads were first checked for the presence of the constant region located in between the stop codon and the barcode; reads lacking this motif were discarded. Barcodes were extracted, corresponding to the 20 nucleotides adjacent to the constant region. Primer sequences were trimmed from all reads. Reads with matching barcodes were pooled, and consensus sequences of the GFP-coding region were obtained independently for the C-terminal and N-terminal halves, as well as for the forward and reverse reads of each half. Barcodes with fewer than five reads in one or both halves were discarded, as well as barcodes with less than 80% agreement for any given position. Consensus sequences of forward and reverse reads were merged, then N- and C-terminal halves were merged to obtain the full coding sequence; barcodes where the expected overlap was less than a 100% match between sequences to be merged were discarded.

Final coding sequences for each barcode were then compared with the wild-type template by global pairwise alignment and mutations were extracted. Coding nucleotide sequences were translated to obtain amino acid sequences.

## Preparation of samples for HiSeq barcode sequencing

Sorted cells were recovered periodically during sorting, and kept on ice to avoid cell division. In order to increase the genetic material available for PCR, recovered cells were plated on LB-zeocin agar and incubated overnight at 37°C. Colonies were pooled and mixed and used as PCR template to amplify the barcode region, as we previously found PCRs directly on the sorted cells to be inefficient.

As with MiSeq sample preparation, two rounds of PCR were performed. The first round consisted of 15 cycles (Encyclo polymerase, Evrogen) and used N-shifted primers to increase complexity, and was gel-purified and used as the template for the second round, which consisted of 9 cycles and added full Illumina TrueSeq adapters. Final products were gel-purified and sent for HiSeq SR100 sequencing. Different Illumina indices were used for different samples, allowing pooling of multiple samples into the same sequencing lane. Sample ratios in each lane corresponded approximately to the numbers of sorted cells. A total of four HiSeq lanes were used, with twelve samples per lane.

## HiSeq data processing

As with MiSeq data, HiSeq sample demultiplexing was performed by the sequencing facility, raw data was converted from.bam to.fastq using Bamtools, and further processing was done with custom Python scripts.

HiSeq reads encompassed the barcode region only: a 20 N primary barcode and a 10 N secondary barcode, with a 10bp-length constant region in between. Reads lacking the constant motif were discarded. Primary and secondary barcodes were extracted from each read, and reads with matching primary barcodes were pooled. If a secondary barcode had fewer than six reads, and differed from another secondary associated to the same primary by two or fewer nucleotides, it was considered to be the product of sequencing errors and its reads were merged together with the more abundant barcode. For each primary-secondary barcode combination, the distribution of reads across the eight green sorting gates was determined.

In order to estimate the actual number of sorted cells from the number of reads, we used 'count control' barcodes mentioned previously: cells of known barcode of which a fixed amount was sorted

into each tube. For each gate, the read counts of all barcodes were normalized according to the average 'count control' read count of that gate.

## Estimating fluorescence for each genotype from HiSeq sequencing of sorted populations

To determine the genotype distribution across brightness populations we used Illumina HiSeq (single-end 100 bp reads). The fluorescence for each genotype was assessed by fitting the calculated number of cells across the sorting gates to the cumulative density function of normal distribution (provided by scipy.stats.norm Python module), taking into account gate border values for every sorter run (*Source data 1*). The fitting was performed with the scipy.optimize.curve_fit Python module, with the initial guess corresponding to the gate with the highest cell count observed. The initial guess for the sigma equaled the width of the gate with maximum cell count. In order to correct for slightly different settings between two FACS machines, the brightness values were matched by linear regression of fluorescence values of known wild-type genotypes.

## Genotype data filtering

Our experimental setup allowed for various sources of replication for each mutant: cells with the same genotype and primary barcode but different secondary barcodes; or the same genotype but different primary barcodes; or the same genotype as well as primary and secondary barcode but sorted on different machines. Such replicates were merged, and assigned a fitness corresponding to the mean of the fitnesses of each individual replicate, weighted by their cell counts (*Source data 2*).

Two sources of internal controls allowed us to check data quality: genotypes corresponding to wild-type proteins, known to be bright, and genotypes corresponding to chromophore-mutated variants known to be dark. Due to the physical limitations of FACS, it is not unexpected for some number of cells to be mis-sorted into the wrong gate, but we expect such events to be associated to low read counts, and for mis-sorted cells to be associated to different fitnesses than correctly sorted cells of the same genotype. Therefore, we discarded nucleotide genotypes with too few replicates, or with too low a cell count, or whose replicates covered too wide a range of measured fitnesses (as determined according to their index of dispersion). Due to differences in library diversity and measurement range between libraries, the particular cutoffs for amacGFP, cgreGFP, and ppluGFP2 differed slightly, and were selected such as to minimize the numbers of false positives and false negatives, while maximizing the total number of retained genotypes. In brief, amacGFP, cgreGFP, and ppluGFP2 genotypes were required to have, respectively: a minimum of 2, 3, and 3 replicates; cell counts over 26, 14, and 23; and indices of dispersion under 525, 575, and 1000. In each case, the final dataset showed no false negatives, that is wild-type proteins measured as dark (based on data from over 1000 nucleotide genotypes with synonymous mutations), while false positive rates, that is genotypes with chromophore mutations measured as having non-zero fluorescence, ranged from 0.24% (ppluGFP2) to 0.47% (amacGFP) to 0.71% (cgreGFP).

Of the surviving nucleotide genotypes, those with synonymous mutations coding for the same protein were merged, and assigned a fitness corresponding to the mean of the fitnesses of the different nucleotide genotypes, weighted by their cell count. The final dataset (*Source data 3*) included 35500, 26165, and 32,260 unique protein sequences respectively for amacGFP, cgreGFP, and ppluGFP2.

## Calculation of epistasis

Our mutant generation strategy created genotypes with an average of 3–4 mutations each. This also led to >1100 single mutants in each gene (*Supplementary file 1*) for which we could directly calculate their individual effects. This allowed us to determine the contribution of epistasis to the fluorescence of genotypes with multiple mutants. Epistasis was calculated as the difference between the measured fluorescence of a genotype and its expected fluorescence under the assumption that the joint effect of multiple mutations is equal to the sum of their individual effects, according to the following equality:

$$epistasis = Effect_{measured} - Effect_{expected} = (F_m - F_{wt}) - \sum (F_i - F_{wt}) x_i$$

where $F_i$, $F_m$ $F_{wt}$ are measured levels of fluorescence of a genotype with mutation $i$, of a genotype $m$ containing one or more mutations, or of the wildtype sequence, respectively, and $x_i = 1$ when mutation $i$ is contained within the genotype $m$ and $x_i = 0$ when it is not. In order to avoid detecting false

epistasis, expected values are capped and cannot be greater than the dataset's maximum observed measurement, nor less than the minimum observed measurement. Instances of genotypes harbouring multiple mutations for which one or more of the mutations has not been observed in isolation were not included in this analysis.

We selected –1 or 1 and –0.3 or 0.3 cutoffs representing strong and weak epistatic interactions, respectively. The fluorescence values used in the study are log10-transformed from the original FACS values. Thus, the –1 or 1 cutoff corresponds to a 10-fold change in fluorescence, while –0.3 or 0.3 is an approximately twofold change. The wildtype sequence measurements fell in a relatively narrow range of fluorescence (*Figure 1c*), with a standard deviation of around 0.03 for three GFP sequences, slightly higher for avGFP (*Table 1*). As the value of ~0.03 was similar regardless of the mean fluorescence, we expect it to hold across other genotypes in the libraries, and would therefore expect that ~99% of measurements should fall within a+/-0.09 interval of the true value (3 standard deviations, which corresponds to a<25% change in fluorescence when converted back to non-log10 values). Both thresholds, 0.3 and 1, for moderate and strong epistasis, respectively, fall comfortably far outside the range of measurement errors caused by experimental noise.

## Protein purification

Wildtype sequences with N-terminal His-tags were cloned into T7 expression vectors via MoClo. Chemically competent BL21-DE3 (New England Biolabs) were transformed and plated on LB agar supplemented with antibiotic and 20 μM IPTG, grown overnight at 30°C and left at room temperature an additional day to allow extensive time for fluorescent protein maturation. Colonies from twenty 12 × 12 cm plates were scraped and recovered in 40 ml of binding buffer (500 mM NaCl, 20 mM Tris-HCl, 25 mM imidazole, pH 8), lysed in a Qsonica Q700 sonicator (20 kHz, amplitude 10, 1 s on/4 s off, 20 min of active sonication time), and centrifuged for 30 min at 20,000 g. The supernatant was recovered and incubated with rotation for 1 hr at 4°C with 3 ml of nickel-sepharose protein purification resin (Cytiva). Before use, resin was washed with 5 volumes of binding buffer, and 5 volumes of distilled water.

After incubation, the protein/resin solution was passed through an empty chromatography column (BioRad Econo-Pac), washed thrice with 20 ml of binding buffer, then protein was recovered in 2–5 ml of elution buffer (500 mM NaCl, 20 mM Tris-HCl, 500 mM imidazole, pH 8).

## Crystallization, data collection, and structure determination

AmacGFP (12 mg/mL in 20 mM Tris-HCl buffer, pH 7.5) was crystallized at 21 °C in 8% PEG 6 K, 3% glycerol, 0.1 M sodium acetate, pH 5.0 supplemented with 5.0% Jeffamine M-600 pH 7.0 according to the Hampton Research Additive Screen protocol using the sitting drop vapor diffusion technique. Crystals grew within 1 week and were flash frozen in liquid nitrogen using mother liquor supplemented with 20% PEG 400 as cryoprotectant.

Diffraction data were collected using the D8 Venture (Bruker AXS, Madison, WI) system that includes an Excillum D2 +MetalJet X-ray source with Helios MX optics providing Ga Kα radiation at a wavelength of 1.3418 Å and a PHOTON III charge-integrating pixel array detector. Data were reduced using Proteum3 software (Bruker AXS). The crystal structures were solved by molecular replacement with MOLREP (*Vagin and Teplyakov, 1997*) using a avGFP mutant as a search model (PDB ID 2AWK). Model building and iterative refinement were performed with Coot (*Emsley and Cowtan, 2004*) and REFMAC (*Murshudov et al., 1997*), respectively. The final statistics of the structure are shown in *Supplementary file 2*. The model has been deposited into the Protein Data Bank (PDB ID 7LG4).

## ΔΔG prediction and residue distance measurements

Calculations were performed using the following structures: avGFP (PDB ID 2WUR), ppluGFP2 (PDB ID 2G3O), cgreGFP (PDB ID 2HPW), and amacGFP (PDB ID 7LG4, this study). For each structure, one (the first) chain was extracted and minimized using Rosetta Relax application (*Nivón et al., 2013*) with constraints to starting coordinates. The total of 50 structures were generated for each protein and the model with the lowest total score was chosen for further calculations. The GFP chromophores' (GYG and SYG) geometries were optimized in Gaussian using density functionals at the B3LYP/6–31++G(d,p) level of theory. The chromophores were treated as non-canonical amino acids (*Renfrew et al., 2012*). ΔΔG calculations (*Source data 4*) were performed for all single mutations except for

nonsense mutations, mutations in the chromophore triade, and positions that are not present in the corresponding crystal structure using Rosetta ddg_monomer application (*Kellogg et al., 2011*). All runs were performed with Rosetta version 3.10. Distances between amino acid residue pairs are available in *Source data 5*.

## Urea sensitivity assays

Absorbance and fluorescence spectra were measured on Biotek SynergyH1 plate readers. For fluorescence, samples consisted of 200 µl of 0.15 µM purified protein in 1 X PBS and either 0 M or 9 M urea, and emission was measured in 5 nm steps from 450 nm to 700 nm upon excitation at 420 nm. For absorbance, samples were identical except for protein concentration, here 18.5 µM, and absorbance was measured in 5 nm steps from 300 nm to 700 nm. In both cases, spectra were continuously measured for around sixty hours, at 42 °C, and a minimum of eight technical replicates were measured for each condition. All plates used were 96-well clear- and flat-bottomed plates; for fluorescence measurements, plates were also black-walled. Blanks containing elution buffer instead of protein (see: Protein purification) were also measured, and their values subtracted from those of the protein samples (*Source data 6*).

To measure refolding kinetics, samples were first denatured by diluting in 9 M urea to a final protein concentration of 0.5 mg/ml and heating at 95 °C for 5 min. 10 µl were then transferred to a 96-well flat-bottomed plate and baseline fluorescence (excitation at 485 nm, emission at 520 nm) was measured on a Biotek SynergyH1 plate reader. A total of 200 µl of 1 X PBS was added via injection and fluorescence was immediately measured for 20 min at intervals of 1 s, or for 13.8 hr at intervals of 50 s.

## Thermosensitivity assays

Thermal unfolding and/or aggregation of purified green fluorescent proteins was monitored by differential scanning fluorimetry (DSF), circular dichroism (CD), and differential scanning calorimetry (DSC), and fluorescence emission during heating was monitored in a Roche Lightcycler 480. Protein samples were diluted in imidazole-free elution buffer (see: Protein purification) from 20 mg/ml stocks in 500 mM imidazole elution buffer. Raw data are available in *Source data 7*.

### Differential scanning fluorimetry

Samples were run in triplicate on a Prometheus NT.48 (NanoTemper Technologies) machine set at 100% excitation power. Samples consisted of 10 µl of 1 mg/ml protein, heated from 20°C to 110°C at a ramp rate of 1 °C per minute; melting temperatures for unfolding and aggregation were determined from the peaks of the first derivatives of either the 350/330 nm emission ratio or the light scattering, respectively. Although all considered GFPs contained a low content of tryptophan, the primary signal source in NanoDSF, all GFPs contained high enough tyrosine content to generate a good signal (avGFP: 1 Trp, 11 Tyr; amacGFP: 1 Trp, 10 Tyr; cgreGFP: 3 Trp, 14 Tyr; ppluGFP2: 0 Trp, 12 Tyr).

### Circular dichroism

Samples consisting of 200 µl of 0.1 mg/ml protein in a 1 mm thickness cuvette were analyzed on a Jasco J-815 CD spectropolarimeter. Initial protein spectra were measured at 30 °C, from 260 nm to 200 nm, and the spectrum of protein-free buffer was subtracted; protein spectra were not measured beyond 200 nm as the high tension voltage in this region increased beyond 700 V, making CD measurements unreliable. The following settings were used for spectra measurements: scanning speed of 100 nm/min; data pitch of 1 nm; digital integration time of 2 s with 1 nm bandwidth; 10 accumulations. After measuring the initial spectra, samples were heated to 98 °C at a rate of 1 °C per minute, and monitored throughout at 218 nm (208 nm in the case of avGFP), a wavelength corresponding to a peak in the spectra. The full spectra were then measured again at 98 °C, under the same settings described above. The single-wavelength melting curves were fitted with a logistic curve, $f\left(x\right) = \frac{L}{1 + e^{-k\left(x - x0\right)}}$, using the Python module scipy.optimize.curve_fit, in order to obtain the melting temperature ($x0$) and the logistic growth rate ($k$).

### Differential scanning calorimetry

One mg/ml protein samples were run in duplicate on a MicroCal PEAQ-DSC (Malvern Panalytical), and measured from 20°C to 110°C at a ramp rate of 1 °C per minute. Melting temperatures (temperature

corresponding to the peak in specific heat capacity or Cp) and enthalpies of denaturation (area under the peak) were determined automatically. DSC runs were performed by the BIC facility of CEITEC MU, Brno.

### Green fluorescence emission upon heating

Fluorescence emission of purified protein samples (0.1 mg/ml, final volume 20 μl in white 96-well plates) during heating from 20°C to 99°C at a ramp rate of ~2 °C/min was measured on a Roche Light-Cycler 480 monitoring the SYBR-Green channel (excitation at 465 nm, collection at 510 nm). (*Source data 7*). Melting temperatures were determined automatically from the melting curve peak.

### SEC-MALS

Size exclusion chromatography/multiangle light scattering analysis was performed on an OmniSEC system (Malvern Panalytical). Samples consisted of 0.2μm-filtered, 1 mg/ml purified proteins in 20 mM Tris pH 8, 150 mM NaCl, 25 mM imidazole buffer. Injection volumes were 50 μl. Samples were measured at 30 °C with a flow rate of 0.7 mL/min. SEC-MALS runs were performed by the BIC facility of CEITEC MU, Brno.

### SDS-PAGE and western blots

Genome-integrated mutant libraries were plated on LB-Zeocin agar plates, colonies were recovered (pelled weight of 0.25 g) and resuspended in 30 mL of lysis buffer (1 X PBS pH 7.4, 150 mM NaCl, supplemented with 50 μl protease inhibitor cocktail (Sigma Aldrich, ref P8340)). Cells were sonicated on a QSonica Q700 (20 kHz, amplitude 10, 1 s on/4 s off, 10 minutes of active sonication). To separate soluble and insoluble fractions, 15 μl of lysate were centrifuged for 10 minutes at 20,000 g, supernatant was collected and the pellet resuspended in 15 μl of lysis buffer. 5 μl of 4 X Laemmli loading dye (BioRad) was added to 15 μl of either total lysate, supernatant, or resuspended pellet. Samples were boiled at 95 °C for 5 min, and run in 4–20% polyacrylamide Mini-Protean precast gels (BioRad) at 100 V for 1 hr. The Protein Precision Plus Standard (BioRad) was used as a molecular weight marker. Gels were stained with a colloidal Coomassie dye, ReadyBlue (Sigma-Aldrich), overnight at room temperature.

For Western blot, gels were transferred to PVDF membranes (BioRad) using a Trans-Blot Turbo Transfer system (BioRad), blocked with EveryBlot blocking buffer (BioRad) for 15 min at room temperature, and incubated overnight at 4 °C with a mouse monoclonal anti-His-tag primary antibody (Abcam, ref ab18184) diluted 1:1000. Membranes were washed in 1 X PBS/0.05% Tween-20 (five 5-min washes), incubated for two hours at room temperature with 1:1000 anti-mouse HRP secondary antibody (Cell Signal, ref #7076), washed, and incubated with SuperSignal West Pico-Plus ECL substrate and imaged on a ChemiDoc MP system (BioRad).

### Experimental testing of predictions

Coding sequences for neural network-generated genotypes (*Source data 8*) were ordered as dsDNA from Twist Biosciences, flanked by BsaI restriction sites for MoClo insertion into destination vectors. GFP sequences were cloned into a medium/low-copy vector conferring zeocin resistance, under a constitutive T5 promoter and lambda t0 terminator, and transformed into XL10-Gold chemically competent cells. Cells were plated on LB-zeocin agar supplemented with ink (1%) to improve fluorescence visualization. Colonies of each construct were picked and sent for Sanger sequence confirmation (Microsynth). Photographs of plates were taken with a Canon EOS 600D SLR camera. For comparison of fluorescence of different genotypes, photographs of plates containing streaks of all wild-types and mutants were photographed under identical conditions (aperture 2.8, ISO 100, 0.8 s exposure time), images were converted to 8 bit in FIJI and median pixel values were determined for each streak. Brightness, contrast, or other image parameters were not altered, and none of the images used contained any saturated pixels.

### Sequence and evolutionary analysis

A total of 68 GFP sequences with confirmed emission in the green spectrum were selected using available information from the literature. These sequences (*Source data 9*) were used in the analysis of the fraction of deleterious amino acids in one of the four wildtype sequences that were neutral in another

genotype (*Figure 5*) and in determining extant amino acid states (*Figure 6—figure supplement 1*). To calculate the rate of evolution of mutations that were used in successful predictions of distant functional GFPs, we aligned these 68 amino acid sequences with muscle (*Edgar, 2004*), trimmed the alignment and made a phylogenetic reconstruction with MrBayes (*Ronquist et al., 2012*), reconstructed the ancestral state of a non-trimmel codon alignments and calculated the ds per each branch of the tree by codeml (*Yang, 2007*). Finally, we compared the rate of evolution between the two amino acid states (the one found in the wildtype and the other corresponding to an amino acid state used in at least one of the successful neural network predictions) to the rate of synonymous evolution (ds) at the same branches (*Source data 10*).

## Modeling the fitness landscape of GFPs with neural networks

For all 4 fitness landscapes, the log10-transformed fluorescence (fluorescence for short) is a bimodal distribution of two normals with very little overlap. One mode corresponds to non-functional genotypes while the other mode corresponds to functional genotypes of near wild-type fluorescence levels. In each dataset, genotypes associated with negative fluorescence have been excluded to ensure that the four distributions cover similar ranges. The genotype-phenotype datasets were split randomly into training, validation and test sets (60%, 20–20%). To evaluate the complexity of the genotype-phenotype relationship in the four landscapes, we trained neural networks of increasing complexity on one-hot encoded protein sequences with the task of predicting fluorescence level. One-hot is a binary encoding that represents which amino acid is present or absent for each position in a sequence. All models were built using Keras (*Chollet, 2015*). Model goodness was measured as the coefficient of determination between known and predicted fluorescence values associated with genotypes in the validation set.

For each dataset, a linear model defined as a neural network containing only an input layer and one layer of a single neuron with linear activation was trained for a maximum of 30 epochs with the objective to minimise the loss as defined by the mean squared error (MSE) between actual and predicted fluorescence levels. Overfitting was prevented by monitoring the validation loss with a patience of 10 epochs. These baseline models output a simple estimate of the fluorescence level, the fitness potential, associated with each genotype. It is simply the summed contribution of mutations assigned individual weights. Models with a sigmoid output node were obtained by adding a single neuron with sigmoid activation function to these architectures and retraining on the training set.

In order to capture and visualise the non-trivial functions transforming the fitness potential into the predicted fluorescence, we trained models containing an input layer, a hidden layer with one linear node, computing the fitness potential, a second hidden layer of 10 sigmoid nodes and one linear output node outputting the fluorescence. The output subnetwork of 10 sigmoids allows the models to approximate a wide variety of sophisticated functions. For each genotype in the validation set, we computed the fitness potential as the output of the first hidden layer, and the predicted fluorescence. The output subnetworks were able to accurately capture the functions transforming the fitness potential into fluorescence level, revealing non-trivial sigmoid functions (*Figure 6—figure supplement 1*).

Optimisation of artificial neural nets was performed using a random grid search approach. Tested architectures contained one input layer, one hidden layer with linear nodes, a second hidden layer with sigmoid nodes, and one linear output node. The two hidden layers were built with random numbers of neurons, picked from 1 to 10, 20, 50, 100, 200. The models also contained one Monte Carlo dropout layer after each hidden layer (rate = 0.1, training = True), but not after the output node (*Figure 6a*). MC Dropout layers present the double advantage of preventing overfitting and allowing the model to predict the fluorescence of each genotype with uncertainty estimates (*Hinton et al., 2012*; *Srivastava et al., 2014*). Each architecture was trained for 10 epochs and the architecture with the smallest loss (MSE) on the validation set was selected for further training to a maximum of 30 epochs. To ensure fair training of the optimised models, the training set was filtered to exclude genotypes containing mutations present in less than 10 distinct genotypes. Since removing a genotype with a rare mutation also decreases the number of occurrences of the other mutations this genotype may contain, the filtration process was repeated until all mutations in the training set were present in at least 10 genotypes and therefore no further genotype had to be removed. To ensure fair scoring of the optimised models, the validation set was filtered to remove genotypes that contained mutations absent from the training set. This ensures the neural networks are trained on enough examples for each mutation, and the model's

final score is not underestimated due to poorly trained mutations present in the validation set. These models were then used as part of a genetic algorithm to predict distant functional genotypes.

For each gene, an additional model with 10, 100 and 1 leaky ReLU nodes was trained and validated independently on 90% and 10% of the dataset respectively for a maximum of 500 epochs, minimising MSE loss. Overfitting was prevented by monitoring the validation loss with a patience of 10 epochs. Coefficients of determination were 0.710, 0.740, and 0.810 for amacGFP, cgreGFP and ppluGFP2, respectively. These models were used to filter genetic algorithm predictions a posteriori with an independent predictor.

## Prediction of distant functional genotypes

Prediction of distant functional genotypes was performed using a genetic algorithm approach. An initial population of 50 wild-type genotypes is initialised. At each generation, the genotypes were shuffled and half of the population was put aside to remain untouched. The other half undergoes crossing-overs and mutations. Crossing-overs were performed randomly along pairs of genotypes without gene conversion. Crossing-overs had a 0.7 probability of occuring in each couple of sequences and the number of crossing-overs was chosen randomly in the range of 0–5. Resulting genotypes (some of which may not have been crossed) underwent a mutagenesis step. Mutations were picked from a random pool containing mutated states but also wild-type states to allow the algorithm a chance to revert evolutionary dead-ends. If the targeted number of mutations defined by the user was exceeded in a genotype, the algorithm removed one previously added mutation from the genotype, allowing heavily mutated sequences to gradually bounce back to the target value. Per amino-acid mutation probabilities were defined empirically in the range of 0.01–0.015. After crossing-overs and mutations, the new genotypes were added to the rest of the population.

Mutations available to the genetic algorithm were selected using the following approach: we excluded mutations that had been seen by the model in less than 10 distinct genotypes during training of the optimised neural nets. From the remaining mutations, we kept those for which we could find in the dataset both genotypes that had those mutations and identical counterparts or 'background' genotypes without these mutations. If at least five pairs of corresponding genotypes could be found, the impact of the mutation was computed by subtracting the fluorescence levels of the genotypes without the mutation to the fluorescence levels of the genotypes that contain the mutation and taking the median. Last, only mutations with a median impact greater or equal to –0.1 log10 fluorescence units were fed into the genetic algorithm. In short, this approach excludes severely deleterious mutations from the genetic algorithm.

Finally, to ensure predicted genotypes did not converge to 'known' functional genotypes, the pool of usable mutations was enriched in mutations that do not lead to states observed in natural GFP sequences. To do so, all sequences were scraped from FPbase (June 2020), filtered to keep green natural ones and aligned on a profile obtained from the wild-type sequence of amacGFP, avGFP, cgreGFP, and ppluGFP2. We then applied in the pool of mutations available to the genetic algorithm, a ratio of 0.6 in favor of mutations that could not be found in natural GFPs.

After the crossing-over and mutation steps, the genotypes were one-hot encoded and their fluorescence level was updated by taking the median of 20 outcomes computed by the optimised neural network. The genotypes were sorted by descending fluorescence and only top genotypes were kept to maintain a constant population size. This process was repeated for several generations. Crossing-over and mutation rates, number of generations, and the ratio between mutated or wild-type available states were adjusted empirically to allow most genotypes in the population to reach the desired number of mutations while evolving to improved fluorescence levels. Notably, the algorithm was stopped a few generations after the median of predicted fluorescence levels in the population reached a plateau. This was to ensure the algorithm selects the best performing genotypes at the desired number of mutations while maintaining sequence diversity in the population. The entire algorithm was repeated until all unique mutations in the pool had been sampled, with a minimum of 10 replicates.

The resulting predictions were filtered to keep unique genotypes that contained the required number of mutations and whose fluorescence ± standard deviation as computed by the optimised and the posteriori neural networks was greater than fluorescence level of their wild-type counterpart. Finally, we used cd-hit (*Fu et al., 2012*; *Li and Godzik, 2006*) iteratively with a similarity threshold

decreasing at each iteration until the remaining set of candidate sequences could be separated into the desired number of clusters. One representative sequence per cluster was picked for experimental validation.

## Software

Visualization of proteins was done in PyMOL (The PyMOL Molecular Graphics System, Version 2.0 Schrödinger, LLC.). Custom scripts for data processing and analysis were written in Python 3, with the most frequently used libraries being NumPy (v. 1.18.1), Pandas (v. 1.3.4), and SciPy (v. 1.7.2). Matplotlib (v. 3.1.3), Seaborn (v. 0.11.2), and cmocean (v. 2.0) were used for data visualization.

## Data availability

All data and programs relevant to our methodology is available at: https://github.com/aequorea238/Orthologous_GFP_Fitness_Peaks (copy archived at swh:1:rev:4bb2791013567442a8ea4d-7735ca79311502bdee; *Somermeyer, 2022*). Cell libraries are available upon reasonable request and subject to a material transfer agreement.

## Acknowledgements

We thank Ondřej Draganov, Rodrigo Redondo, Bor Kavčič, Mia Juračić and Andrea Pauli for discussion and technical advice. We thank Anita Testa Salmazo for advice on resin protein purification, Dmitry Bolotin and the Milaboratory (milaboratory.com) for access to computing and storage infrastructure, and Josef Houser and Eva Fujdiarova for technical assistance and data interpretation. Core facility Biomolecular Interactions and Crystallization of CEITEC Masaryk University is gratefully acknowledged for the obtaining of the scientific data presented in this paper. This research was supported by the Scientific Service Units (SSU) of IST-Austria through resources provided by the Bioimaging Facility (BIF), and the Life Science Facility (LSF). MiSeq and HiSeq NGS sequencing was performed by the Next Generation Sequencing Facility at Vienna BioCenter Core Facilities (VBCF), member of the Vienna BioCenter (VBC), Austria. FACS was performed at the BioOptics Facility of the Institute of Molecular Pathology (IMP), Austria. We also thank the Biomolecular Crystallography Facility in the Vanderbilt University Center for Structural Biology. We are grateful to Joel M Harp for help with X-ray data collection. This work was supported by the ERC Consolidator grant to FAK (771209—CharFL). KSS acknowledges support by President's Grant $\text{МК}$–5405.2021.1.4, the Imperial College Research Fellowship and the MRC London Institute of Medical Sciences (UKRI MC-A658-5QEA0). AF is supported by the Marie Skłodowska-Curie Fellowship (H2020-MSCA-IF-2019, Grant Agreement No. 898203, Project acronym "FLINDIP"). Experiments were partially carried out using equipment provided by the Institute of Bioorganic Chemistry of the Russian Academy of Sciences $\text{С}$ore Facility (CKP IBCH). This work was supported by a Russian Science Foundation grant 19-74-10102. This project has received funding from the European Union's Horizon 2020 research and innovation programme under the Marie Skłodowska-Curie Grant Agreement No. 665,385.

## Additional information

### Funding

| Funder | Grant reference number | Author |
| --- | --- | --- |
| European Research Council | 771209-CharFL | Fyodor A Kondrashov |
| MRC London Institute of Medical Sciences | UKRI MC-A658-5QEA0 | Karen S Sarkisyan |
| President's Grant | $\text{МК}$-5405.2021.1.4 | Karen S Sarkisyan |
| Marie Skłodowska-Curie Fellowship | 898203 | Aubin Fleiss |
| Russian Science Foundation | 19-74-10102 | Karen S Sarkisyan |

| Funder | Grant reference number | Author |
| --- | --- | --- |
| Marie Skłodowska-Curie Grant Agreement | 665385 | Louisa Gonzalez Somermeyer |
| FWF Austrian Science Fund | I5127-B | Fyodor A Kondrashov |

The funders had no role in study design, data collection and interpretation, or the decision to submit the work for publication.

## Author contributions

Louisa Gonzalez Somermeyer, Conceptualization, Data curation, Formal analysis, Investigation, Methodology, Project administration, Validation, Visualization, Writing – original draft; Aubin Fleiss, Conceptualization, Formal analysis, Investigation, Software, Validation, Visualization, Writing – original draft; Alexander S Mishin, Conceptualization, Formal analysis, Investigation, Supervision, Validation; Nina G Bozhanova, Conceptualization, Formal analysis, Investigation, Methodology, Resources, Validation, Visualization, Writing – original draft; Anna A Igolkina, Formal analysis, Investigation, Methodology; Jens Meiler, Funding acquisition, Methodology, Project administration, Resources, Supervision; Maria-Elisenda Alaball Pujol, Formal analysis, Investigation; Ekaterina V Putintseva, Conceptualization, Formal analysis, Investigation, Methodology, Supervision, Visualization; Karen S Sarkisyan, Conceptualization, Formal analysis, Funding acquisition, Investigation, Methodology, Project administration, Resources, Software, Supervision, Validation, Visualization; Fyodor A Kondrashov, Conceptualization, Formal analysis, Funding acquisition, Investigation, Methodology, Project administration, Resources, Supervision, Validation, Visualization, Writing – original draft

## Author ORCIDs

Louisa Gonzalez Somermeyer (iD) http://orcid.org/0000-0001-9139-5383
Alexander S Mishin (iD) http://orcid.org/0000-0002-4935-7030
Nina G Bozhanova (iD) http://orcid.org/0000-0002-2164-5698
Anna A Igolkina (iD) http://orcid.org/0000-0001-8851-9621
Jens Meiler (iD) http://orcid.org/0000-0001-8945-193X
Maria-Elisenda Alaball Pujol (iD) http://orcid.org/0000-0002-1868-2674
Karen S Sarkisyan (iD) http://orcid.org/0000-0002-5375-6341
Fyodor A Kondrashov (iD) http://orcid.org/0000-0001-8243-4694

## Decision letter and Author response

Decision letter https://doi.org/10.7554/eLife.75842.sa1
Author response https://doi.org/10.7554/eLife.75842.sa2

---

# Additional files

## Supplementary files

• Transparent reporting form

• Supplementary file 1. Selected statistics of genotypes at different divergence from five GFP sequences.

• Supplementary file 2. Data Collection and Refinement Statistics.

• Source data 1. Absolute values for the borders between gates in the green channel during sorting, for all genes and machines, and the corrections applied to match values between the machines.

• Source data 2. Dataframes containing the distribution across gates of all primary-secondary barcode combinations, along with their fitted fitness values (see Materials and methods). Data are not filtered according to cell count, number of replicates, etc. One dataframe per gene and machine.

• Source data 3. Dataframes linking nucleotide or protein genotypes to their measured fluorescence level (see Materials and methods). Mutations in genotypes are labeled in the format AiB, where A is the original wildtype state, B is the mutated state, and i is the position (counting starts from Methionine = 0). In the nucleotide dataset, 'n_replicates' refers to the combined number of distinct barcodes representing a genotype and machines it was measured on. In the amino acid dataset, 'n_replicates' refers to the number of synonymous nucleotide sequences

measured for each protein sequence. Nucleotide genotypes and amino acid genotypes are on separate tabs in the file.

• Source data 4. Table containing ddG predictions for single mutations in avGFP, amacGFP, amacGFP:V12L, cgreGFP, and ppluGFP2. Residue positions are labeled starting from 0 (methionine).

• Source data 5. Dataframes containing the minimum physical distance between pairs of residues inside the 3D GFP structures, in Angstroms. Row and column indices represent the residue position within the protein, starting from 0 for the initial methionine. Matrices for different proteins are included in different tabs in the file.

• Source data 6. Table containing absorbance values (from 300 to 700nm) and fluorescence emission values (from 450nm to 700nm, upon 420nm excitation) for all genes, in 9M urea and PBS, measured on a plate reader at multiple consecutive time points. Blank control values are already subtracted. Absorbance and fluorescence data are listed on separate tabs

• Source data 7. Raw data from differential scanning fluorimetry and calorimetry, circular dichroism, and qPCR melting curves.

• Source data 8. Coding sequences for neural network-generated genotypes, and their predicted and observed levels of fluorescence.

• Source data 9. Table of over 70 documented natural fluorescent proteins used during analyses, including name, species, sequence, original reference and, where possible, accession numbers and measured excitation/emission peaks.

• Source data 10. Estimated rates of evolution of amino acid states used in prediction of novel GFP sequences on each branch of the phylogeny of extant GFPs.

### Data availability

All data generated or analysed during this study are included in the manuscript and supporting file and are available on GitHub https://github.com/aequorea238/Orthologous_GFP_Fitness_Peaks.

The following dataset was generated:

| Author(s) | Year | Dataset title | Dataset URL | Database and Identifier |
| --- | --- | --- | --- | --- |
| Somermeyer LG | 2022 | Heterogeneity of the GFP fitness landscape and data-driven protein design | https://github.com/aequorea238/Orthologous_GFP_Fitness_Peaks | GitHub, Fitness_Peaks |

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
