## [Editor Report]

Using high throughput mutagenesis, this work shows that evolutionary distance between homologous genes is not predictive of how these genes' functions will change in response to similar mutations. This suggests that the starting gene sequence will influence how the synthetic design of new protein functions can occur and also supports a role for conditionality in the natural evolution of protein functions.

---

## [Decision Letter]

**Decision letter after peer review:**

Thank you for submitting your article "Heterogeneity of the GFP fitness landscape and data-driven protein design" for consideration by *eLife*. Your article has been reviewed by 3 peer reviewers, one of whom is a member of our Board of Reviewing Editors, and the evaluation has been overseen by Naama Barkai as the Senior Editor. The reviewers have opted to remain anonymous.

Essential revisions:

1. More clarity in what is and what is not new with more caution on the claims.

2. Support on the choice of a single absolute epistatic threshold for all tests.

3. How much of the total amino acid space was tested for each protein considering the median level of non-synonymous changes?

*Reviewer #1 (Recommendations for the authors):*

The authors compare different fluorescence proteins for their mutational spectrum to compare the differential sensitivity of the proteins. This shows that the four proteins even though structurally similar have very different mutational spectra both for additive and epistatic interactions.

The main conclusion is interesting but it has a cold absence of discussion about the host organisms and treats the four proteins as almost random samples of this protein space. However, is it possible that the natural differences in the environment of these four species may have shaped these proteins by selection to different temps, temp variability that could be influencing the results such that there is an underlying basis for why these proteins may not behave as expected?

I understand the intent of the graphical representation in Figure 1A and the presentation of sharp vs flat peaks. But is it correct to imply that the valleys in between these peaks have been sampled to ascertain if there are unoccupied peaks? Presently the conceptual representation does give an impression of an absence of any other peaks in this genetic space which I am not sure is supported. This comes from the fact that the space around peak is only sampled using 3-4 amino acid substitutions. Admittedly it is not currently possible to simply test the entire space, I'm simply asking if there is a way to show this unknown possibility within the conceptual figure.

I may have missed this but is there a way to estimate false negative rates on epistasis when comparing the four proteins? The mutational analysis is random so I would presume that there is some imbalance in sampling which could create a false negative issue when comparing between proteins. I'm presuming that this is not enough to change the pattern but would help to clarify.

Are Figure 3a and 3b swapped? In the text, 3a is about neutrality while it seems to be 3b in the figure by the y-axis label.

*Reviewer #2 (Recommendations for the authors):*

I have summarized some of my concerns below and hope that revising the manuscript improve readability and clarity of the work.

FACS sorting condition – Figure S2 showed their experimental conditions for the gating in FACS. I feel odd that they put horizontal gating when you see the simple tendency in diagonal. It is very common to see such diagonal line as it reflects the noise of expression (or cell size). Then putting a gate in horizontal manner essentially make some bias in sampling. It might be ok, but did the authors ensured their screening/deep mutational scanning strategy by comparing isolated variants (I don't see any figure in this paper)?

The data presentation in Figure 2. Figure 2A and 2C are very misleading as it looks like almost no variant lost fitness up to 2-3 mutations while the distribution of single mutational effects (Figure 2) clearly showed some fraction of mutations killed the protein. I am not sure why the authors decided to generate figures based on the median (figure 2a) or "representative median at least 15 available genotypes (figure 2c)". It is clearly biasing the data presentation toward what the authors want to claim "much difference in mutational robustness". Figure S3-b is far better presentation as it is transparent and actually much more understandable. They should remove Figure 2a and 2c and replaced by Figure S3-b

Related to the above, I do not understand Figure 2d either. Figure S3-b showed the accumulation of 7 mutations looks to me about 50% of variants are dead for amacGFP, while the line in Figure 2d showed more than 90% genotypes are functional. Is that they calculated from single point mutations? I am not sure why this is needed when they have the data. Even if they calculated from single point mutations, about 5% of single point mutations are dead (Table S1), so the accumulation of 7 should read to only about 70% variants are functional (0.95^7^)? Anyway, I believe that FigS3-b showed the all information. Figures 2d seem to be just misleading figures.

I certainly appreciate the efforts that the authors put to measure various biophysical parameters, and tried to correlate them to the mutational robustness. Regardless much effort, they did not see the correlation and they have spent quite substantial arguments for the reasons why they did not see a clear correlation. This certainly reflect the complication of protein folding and measuring the effect of mutations on protein folding. A conventional theory of protein stability and mutational robustness is too simplified and people use this argument tend to ignore the fact that thermostability (of the folded state) does not reflect "folding ability" of each protein in the cell. Moreover, what the authors missed to discuss is that, as far as I understand, GFPs generates the chromophore when they are translated and folded in the cell. So the denatured GFP molecules are different from newly translated polypeptide and the biophysical and biochemical parameters obtained from refolding experiments of GFP may not even reflect the folding process in the cell. Also it is known that refolding of GFP is very slow as they showed in Figure S7 (it takes over hour in the test tube), suggesting refolding (and folding) process of GFPs are highly complicated. Thus it is not surprising at all that the authors did not see any correlation between the thermostability/refolding rate and mutational robustness.

While the authors claimed that it is surprising that the sequence distance does not correlate with the mutational robustness, I don't think it is the case anymore based on what we already know from previous literature. For example, it has been shown that (e.g., Bloom et al., PNAS 2005 PMID 15644440, similar work by Bershtein JMB 2008 PMID 18495157) single point mutation can alter the robustness to mutations of a protein. Moreover, it is general knowledge that most mutations affect protein stability (PMID: 17482644) and the accumulation of multiple mutations would expect to alter the stability (or foldability) of the protein substantially. Thus, It would be much more common to think that four distinct sequences (at least more than 40 mutations separate each other) would exhibit distinct behaviours depending on each sequence's property.

Epistasis calculation – the authors have calculated epistasis and decided '1" is the cutoff for significant and nonsignificant epistasis. What is the bases for the decision? I don't see anything in the main text and may be hidden in the methods. But the justification should be clearly described in the main text based on their experimental noise as it looks to me affecting the results (Figure S5) significantly when they use different cut-offs.

The data presented in Figure 3 is very hard to understand how the authors calculated. As their mutational design is based on error-prone mutations, they do not cover all possible mutations. Also each GFP templates exhibit different WT sequences, there must be only a handful of mutations were commonly observed among the four different experiments. It is not clear how they calculate those numbers in Figure 3, and they should provide much more detailed explanations for this. e.g., How many identical mutations are observed out of X all possible mutations? How they calculate epistasis when the wild type sequence is different? For example, in a 81% diverged pair, only 19% of the positions are calculated? How many mutations actually are observed in both templates? Then 4% out of 19% of the positions exhibit epistasis (figure a – that means only 2 positions showed epistasis)? In any case, more detailed numbers and data processing should be described in the methods as well as in the main text.

While it is interesting that ML can successfully predicted functional variants from a highly fragile GFP (cgreGFP), the authors did not provide much insights into the details. The authors mentioned that the deep mutational scanning data captured negatively epistatic pairs and ML avoided for the prediction. Is that something that the authors can dig and present as data? Currently, it is just a general interpretation and not beyond a hypothesis. In Figure S11, the authors presented epistasis between mutations, and thus they should be able to comment about the designed sequences, e.g., how ML predicted sequences are recapitulated the sequence space around each GFP. All designed sequences just eliminated negative epistatic combinations or even identified any positive epistasis that can compensate each other. Also, it was not clear to me that they only used amino acid mutations that were observed in either phylogeny or their deep mutational scanning dataset (for each GFP)? Or they went to other amino acids that they experimentally did not observe.

*Reviewer #3 (Recommendations for the authors):*

In Figure 3, the description of panels A and B appears to be swapped with respect to the Y axis labels.

The results shown in Figure S5A are unexpected. Taking amacGFP as an example, the graph shows that genotypes with two large-effect mutations, shown in the bottom left corner of this graph, typically show no epistasis (yellow points). According to equation 1, this indicates that the fluorescence of such genotypes should be equal to the sum of effects of the two mutations, i.e. close to -2.4 (-1.2 + -1.2). However, fluorescence = -2.4 appears to be lower than the dynamic range of fluorescence for amacGFP (see for example Figure 1C, where the fluorescence effect of all genotypes appears to be in the range from 0 to -1.5). Can you clarify this? Perhaps a non-additive model of epistasis was used in this figure?

Figure S9 legend: amacGFP:V14L should be V12L.

Figure S9: why were ddG predictions excluded for proline and glycine mutations?

It would be good to discuss the generality of these results. The fitness landscapes of GFP homologues are bimodal, with mutations tending to have either very little effect, or to cause large reduction of fluorescence. Is that a common property of fitness landscapes, or is it specific to GFP, or to the experimental technique used to measure fitness?

Can the data be used to discriminate between second and higher-order epistasis?

---

## [Author Response]

Essential Revisions (for the authors):1. More clarity in what is and what is not new with more caution on the claims.

To improve clarity and to ensure appropriately cautious note with our claims we have implemented the following changes.

a) We have rewritten the discussion to note the possibility that heterogeneity of fitness shape may be related to the environment that the species typically inhabit. We also put this possibility into a greater context of selection on mutational robustness.

b) Clarified the conceptual nature of Figure 1a in the figure legend.

c) As asked by Referee 2, we have replaced Figure 2a (Figure 3a now) by what used to be Figure S3b.

d) We clarified our meaning in figure legends of Figure 2c and Figure 2d (they are Figure 3c and 3d in the revision).

e) In the main text we now mention chromophore maturation as a possible factor in the thermostability and fluorescence relationship.

f) We added a discussion of the evolutionary expectations that arise from the observation that mutational robustness may be a selectable trait.

g) We added information on the number of amino acid states shared between the pairwise comparisons in Figure 3 (Figure 5 in the revision), and rewritten the associated main text and the figure legend to make the meaning of the figure clearer.

h) We now mention in the main text that our Machine Learning algorithm used only those amino acid states that were seen in our data and for experimental verification we selected those sequences that had the most amino acid states that were not observed in any wildtype GFP sequence available in GenBank. The exact numbers are available in Supplementary Data 8, which is also now mentioned in the main text.

i) We rewrote the figure legend of Supplementary Figure 5 (Figure 3—figure supplement 1 in the revision) to make its meaning clearer.

j) We deleted supplementary tables S3 (genetic construct sequences) and S4 (primer sequences) and integrated them into the Key Resources Table, and recoloured many of the figures to make colour schemes more consistent and visually intuitive.

k) We have gone through the entire manuscript and softened any claims that may be construed as lacking an abundance of caution. Specifically, we deleted the word “counterintuitively” from the abstract, replaced “striking” with “notable”, “surprisingly” with “contrary to our expectation” in the main text and removed the characterization of the observed heterogeneity of the fitness peaks as “unexpected”. We believe that in the present text we do not make any strong unsubstantiated claims or claim novelty where none is found. If there are other parts of the manuscript that the referees or editors feel we need to modify, we hope that they will be pointed out.

2. Support on the choice of a single absolute epistatic threshold for all tests.

We now include a new paragraph in the section “Calculation of epistasis” that describes our rationale for selecting the epistasis thresholds of a 10-fold and 2-fold difference. These thresholds are only applied to data presented in Table 1, and Figure 3—figure supplement 1 and Figure 4—figure supplement 5. All other data, figures or tables are not subject to these thresholds.

3. How much of the total amino acid space was tested for each protein considering the median level of non-synonymous changes?

We have now expanded Supplementary Table 1 to include the count of the fraction of amino acid spaces tested as a function of the mutational distance from each of the four GFP sequences.

Reviewer #1 (Recommendations for the authors):The authors compare different fluorescence proteins for their mutational spectrum to compare the differential sensitivity of the proteins. This shows that the four proteins even though structurally similar have very different mutational spectra both for additive and epistatic interactions.The main conclusion is interesting but it has a cold absence of discussion about the host organisms and treats the four proteins as almost random samples of this protein space. However, is it possible that the natural differences in the environment of these four species may have shaped these proteins by selection to different temps, temp variability that could be influencing the results such that there is an underlying basis for why these proteins may not behave as expected?

The referee is correct, that the assayed proteins come from different species and, therefore, may not be equivalent with regard to what environment they are adapted to. Indeed, among the environmental influences may be temperature, salinity, intracellular environment, interaction with other proteins, etc, etc. Without an experimental assay of how the mutants behave in their natural environment it is not possible to formulate a good answer about which of these factors may be important. However, we test fluorescence of the mutants of all four proteins in the same environment, which is the cell of *E. coli*, which is, at least phylogenetically, equidistant to all of the four species and all of our conclusions are specific to the in vitro environment in which we test. We now add this point in the discussion.

I understand the intent of the graphical representation in Figure 1A and the presentation of sharp vs flat peaks. But is it correct to imply that the valleys in between these peaks have been sampled to ascertain if there are unoccupied peaks? Presently the conceptual representation does give an impression of an absence of any other peaks in this genetic space which I am not sure is supported. This comes from the fact that the space around peak is only sampled using 3-4 amino acid substitutions. Admittedly it is not currently possible to simply test the entire space, I'm simply asking if there is a way to show this unknown possibility within the conceptual figure.

The referee is correct, on the scale of sequence divergence this figure aims to represent there are undoubtedly many other functional GFP sequences we have not assayed. There could literally be trillions of them, or maybe just a few hundred, we simply have no idea how many. Thus, instead of trying to represent this unknown value in a conceptual figure we have now expanded the description of Figure 1a to make sure that the readers understand this point.

I may have missed this but is there a way to estimate false negative rates on epistasis when comparing the four proteins? The mutational analysis is random so I would presume that there is some imbalance in sampling which could create a false negative issue when comparing between proteins. I'm presuming that this is not enough to change the pattern but would help to clarify.

We are not entirely sure what the referee is asking here. We do report the false negative (and false positive) rates of observations specific to each protein we assayed (Table 1), i.e. control genotypes whose functionality is known but are classified as non-functional in our data, and vice versa. As epistasis is defined as the difference between the expected and observed values, any accuracy of estimation of epistasis follows from accuracy of the fluorescence measurements of the genotypes in question. As error rates among the control barcodes are low (Table 1), we do not expect a significant underestimation of the pervasiveness of epistasis. The previous study of the avGFP landscape (Sarkisyan et al., 2016) did include an estimation of false epistasis discovery due to data noise and found it to be negligible.

Having said that, the only analysis reported in our manuscript that compares the impact of the same mutations in different proteins is in Figure 3 (Figure 5 in the revision). For that analysis, when comparing two proteins we considered only those amino acid states that were observed in both sequences. The false negative rate for this analysis, however, is derived in the same manner as for all others. For increased clarity, we now added the number of amino acid substitutions that were considered in each pairwise comparison.

Are Figure 3a and 3b swapped? In the text, 3a is about neutrality while it seems to be 3b in the figure by the y-axis label.

We thank the referee for catching this. Panels a and b in Figure 3 (Figure 5 in the revision) are now labelled correctly.

Reviewer #2 (Recommendations for the authors):I have summarized some of my concerns below and hope that revising the manuscript improve readability and clarity of the work.FACS sorting condition – Figure S2 showed their experimental conditions for the gating in FACS. I feel odd that they put horizontal gating when you see the simple tendency in diagonal. It is very common to see such diagonal line as it reflects the noise of expression (or cell size). Then putting a gate in horizontal manner essentially make some bias in sampling. It might be ok, but did the authors ensured their screening/deep mutational scanning strategy by comparing isolated variants (I don't see any figure in this paper)?

The reviewer is correct in noting the diagonal tendency in the FACS data. This is expected, as the mKate2-GFP fusion protein setup necessarily results in a 1:1 ratio of mKate2:GFP, making the red and green signals directly proportional to each other for any given functional variant. The possibility of selecting diagonal gates to accommodate this was indeed considered and discussed with FACS facility staff at the time, but unfortunately there appeared to be no straightforward way of setting gates in this manner -- at least, for the FACS Aria III / Diva software -- short of manually drawing and lining up polygons side by side, which would be an extra source of human error as well as potentially lead to the sampling of cells with different expression levels. We expect that the narrowness of the selected red gate, as well as use of multiple barcode replicates for the same genotype, should minimise any noise due to this.

We have not compared the fluorescence of isolated variants to our high throughput screening data in this work, however, it was done in Sarkisyan et al., (2016) for avGFP, so we felt it is not necessary to demonstrate this again for the three new proteins.

The data presentation in Figure 2. Figure 2A and 2C are very misleading as it looks like almost no variant lost fitness up to 2-3 mutations while the distribution of single mutational effects (Figure 2) clearly showed some fraction of mutations killed the protein. I am not sure why the authors decided to generate figures based on the median (figure 2a) or "representative median at least 15 available genotypes (figure 2c)". It is clearly biasing the data presentation toward what the authors want to claim "much difference in mutational robustness". Figure S3-b is far better presentation as it is transparent and actually much more understandable. They should remove Figure 2a and 2c and replaced by Figure S3-b

We have now replaced Figure 2a (Figure 3a in the revision) with Figure S3b. We have chosen to keep Figure 2c (Figure 3c in the revision) because it is displaying a very different representation of the fitness landscapes of these four proteins not found in any other figure – it is the median fluorescence of genotypes derived not from the wildtype sequences but from sequences one mutation away from the wildtype. We, therefore, kept Figure 2c (Figure 3c in the revision) and we have rewritten the figure legend for Figure 2c (Figure 3c in the revision) to make sure that it is clearer exactly what it represents.

Related to the above, I do not understand Figure 2d either. Figure S3-b showed the accumulation of 7 mutations looks to me about 50% of variants are dead for amacGFP, while the line in Figure 2d showed more than 90% genotypes are functional. Is that they calculated from single point mutations? I am not sure why this is needed when they have the data. Even if they calculated from single point mutations, about 5% of single point mutations are dead (Table S1), so the accumulation of 7 should read to only about 70% variants are functional (0.95^7^)? Anyway, I believe that FigS3-b showed the all information. Figures 2d seem to be just misleading figures.

The referee is correct, if 5% of single mutants are dead then genotypes with 7 mutations should be, on average, 70% dead (0.95^7^). However, if we observe that 90% of genotypes with 7 mutations are dead we can infer that for 20% of genotypes their low fluorescence has to be explained by epistatic interactions. The data reported in Figure 2d (Figure 3d in the revision) is actually more specific than that: for each genotype with >1 mutation we calculate if it is expected to be functional if the mutations in that genotype were non-epistatic and then we compare this estimate to its observed fluorescence. Thus, the figure shows what fraction of genotypes require epistasis to explain its fluorescence level. We have now rewritten the figure legend to make this point clearer. We have also changed the Y-axis label to make sure that the readers do not mistake it for the actual fraction of functional genotypes (it is the fraction of functional genotypes without epistasis).

I certainly appreciate the efforts that the authors put to measure various biophysical parameters, and tried to correlate them to the mutational robustness. Regardless much effort, they did not see the correlation and they have spent quite substantial arguments for the reasons why they did not see a clear correlation. This certainly reflect the complication of protein folding and measuring the effect of mutations on protein folding. A conventional theory of protein stability and mutational robustness is too simplified and people use this argument tend to ignore the fact that thermostability (of the folded state) does not reflect "folding ability" of each protein in the cell. Moreover, what the authors missed to discuss is that, as far as I understand, GFPs generates the chromophore when they are translated and folded in the cell. So the denatured GFP molecules are different from newly translated polypeptide and the biophysical and biochemical parameters obtained from refolding experiments of GFP may not even reflect the folding process in the cell. Also it is known that refolding of GFP is very slow as they showed in Figure S7 (it takes over hour in the test tube), suggesting refolding (and folding) process of GFPs are highly complicated. Thus it is not surprising at all that the authors did not see any correlation between the thermostability/refolding rate and mutational robustness.

We agree with the referee that there could be many reasons why the complexity of protein folding may not be reflected in the measured effect of mutations on fluorescence. As the referee suggested, in the main text we have now added a statement about the possibility of the impact of GFP folding on chromophore maturation as one of the factors that may be influencing its relationship between fluorescence and thermostability. We also now mention in the main text that our measurements of GFP thermostability in vitro may not necessarily be identical to the folding ability of natively folded GFPs.

While the authors claimed that it is surprising that the sequence distance does not correlate with the mutational robustness, I don't think it is the case anymore based on what we already know from previous literature. For example, it has been shown that (e.g., Bloom et al., PNAS 2005 PMID 15644440, similar work by Bershtein JMB 2008 PMID 18495157) single point mutation can alter the robustness to mutations of a protein. Moreover, it is general knowledge that most mutations affect protein stability (PMID: 17482644) and the accumulation of multiple mutations would expect to alter the stability (or foldability) of the protein substantially. Thus, It would be much more common to think that four distinct sequences (at least more than 40 mutations separate each other) would exhibit distinct behaviours depending on each sequence's property.

We disagree with the referee’s logic here. Undoubtedly a single mutation can reduce robustness (the Bloom et al., 2005 and Bernstein et al., 2008 are examples of this) but it does not follow that robustness as a trait is not expected to be conserved. For example, most mutations in an enzyme are expected to alter its catalytic activity but the null expectation is that similar enzymes have similar function. The reason why we expect a correlation between enzymatic function and sequence divergence is because function is under negative selection and the abundant mutations that change the functional properties of the enzyme are eliminated by selection. It has been recently claimed that robustness of GFP is exactly such a trait under selection (Zheng et al., Science, 2020) leading to the null hypothesis that the degree of robustness as a selected trait must also correlate with sequence divergence. We have now modified the manuscript to better reflect this logic and removed the word “unexpected” from the discussion.

Epistasis calculation – the authors have calculated epistasis and decided '1" is the cutoff for significant and nonsignificant epistasis. What is the bases for the decision? I don't see anything in the main text and may be hidden in the methods. But the justification should be clearly described in the main text based on their experimental noise as it looks to me affecting the results (Figure S5) significantly when they use different cut-offs.

Indeed, the cut-offs are somewhat arbitrary. They have been selected for arithmetic simplicity and to ensure a low false positive and negative rates. The fluorescence values used in the study are log10-transformed from the original FACS values. An effect of -1 or 1 therefore indicates a 10-fold change in fluorescence, while -0.3 or 0.3 indicates an approximately 2-fold change. The wildtype sequence measurements fell in a relatively narrow range (Figure 1c), with a standard deviation of around 0.03 for all genes, slightly higher for avGFP that was measured before with a slightly different setup (Table 1). As the value of ~0.03 was similar regardless of the mean fluorescence, we expect it to hold across other genotypes in the libraries, and would therefore expect that ~99% of measurements should fall within a +/- 0.09 interval of the true value (3 standard deviations, which corresponds to a <25% change in fluorescence when converted back to non-log10 values). Both thresholds, 0.3 and 1, for moderate and strong epistasis, respectively, fall comfortably far outside the range of measurement errors caused by experimental noise. Additionally, we opted for the 0.3 cut-off because 2-fold-changes is a common cutoff in a variety of fields (and it was used by Sarkisyan et al., 2016).

These cutoffs are only used to report data in two rows of Table 1 and Figure 3—figure supplement 1 and Figure 4—figure supplement 5. Thus, we thought it overkill to add this discussion in the main text, but we expanded the supplementary methods section “Calculation of epistasis” to include these arguments there.

The data presented in Figure 3 is very hard to understand how the authors calculated. As their mutational design is based on error-prone mutations, they do not cover all possible mutations. Also each GFP templates exhibit different WT sequences, there must be only a handful of mutations were commonly observed among the four different experiments. It is not clear how they calculate those numbers in Figure 3, and they should provide much more detailed explanations for this. e.g., How many identical mutations are observed out of X all possible mutations? How they calculate epistasis when the wild type sequence is different? For example, in a 81% diverged pair, only 19% of the positions are calculated? How many mutations actually are observed in both templates? Then 4% out of 19% of the positions exhibit epistasis (figure a – that means only 2 positions showed epistasis)? In any case, more detailed numbers and data processing should be described in the methods as well as in the main text.

Figure 3 (Figure 5 in the revision) answers two rather straightforward questions. Figure 3a (Figure 5a in the revision) asks “out of all mutations in our dataset that are neutral in one GFP sequence, what fraction are deleterious in another GFP sequence?”. Figure 3b (Figure 5b in the revision )asks “out of all pairs of sites that interact epistatically in one GFP sequence, what fraction also interacts epistatically in another GFP sequence?”. We have now added the numbers of observations on which this is based and rewritten the figure legend to make it clearer.

The reviewer is correct that we do not cover all possible mutations at all sites. However, the effects of over 1000 single mutations (>90%) were measured in each of the libraries (reported in Table S1). Several hundred specific mutations were shared between any given pair of genes, meaning that we observed variants of both genes where the same (structurally aligned) position was mutated to the same final amino acid state (regardless of the initial, wildtype state at that position). We have added the exact number of amino acid states considered in each pairwise protein comparison of neutrality change in Figure 3a (Figure 5a in the revision), and clarified the figure legend to reflect this. Only genotypes with a single mutation were considered in this analysis.

The calculation of epistasis is not dependent on the template sequence. The presence of several thousand double-mutant genotypes in all libraries (Table S1) allowed us to estimate epistasis (or lack thereof) between many pairs of sites within the same gene, leading to a list of site pairs where epistasis was detected in each gene. We then compare these lists of sites and their overlap across genes. In sum, while Figure 3a (Figure 5a in the revision) is about the change in effect of single mutations as a function of the background they occur in, Figure 3b (Figure 5b in the revision) is about whether the spatial positions of epistatic sites are maintained in different backgrounds.

While it is interesting that ML can successfully predicted functional variants from a highly fragile GFP (cgreGFP), the authors did not provide much insights into the details. The authors mentioned that the deep mutational scanning data captured negatively epistatic pairs and ML avoided for the prediction. Is that something that the authors can dig and present as data? Currently, it is just a general interpretation and not beyond a hypothesis. In Figure S11, the authors presented epistasis between mutations, and thus they should be able to comment about the designed sequences, e.g., how ML predicted sequences are recapitulated the sequence space around each GFP. All designed sequences just eliminated negative epistatic combinations or even identified any positive epistasis that can compensate each other.

To answer this question, we will take a few steps back to discuss what machine learning does. Generally, machine learning methods are used to approximate the underlying mathematical function that describes the relationship between the input and the output. In our case, the input are the protein sequences and the output are the fluorescence intensities of those proteins.

The way these functions are approximated is through iterative comparison of the predicted and true values and tweaking the parameters, so that these values get more similar. The higher the complexity of the underlying function is, the more time, effort and data it takes to accurately approximate it. In the simplest case scenario, if there was no epistasis at all, we would have been able to describe the sequence-to-function relationship with a linear function. In that case, during the training phase the weight of each mutation would have been iteratively optimised, so that the weighted sum of all of the observed mutations is as close to the true fluorescence value as possible. It would have been easy to interpret the results of such a linear function optimisation, because all we would need to do would be to look at the assigned weights and see which mutations were beneficial, deleterious or neutral.

However, the higher the complexity of the underlying sequence-to-function is, the harder it is to interpret its meaning. In our case, the underlying mathematical function describing the sequence-to-function relationship of cgreGFP had ~200 dimensions (as opposed to a linear function, which is two-dimensional). At the same time, comparing the fluorescence values predicted by our model with the values obtained assuming no epistatic interactions at all, we can conclude that the mathematical approximation of the sequence-to-fluorescence relationship we have identified does account for the interactions between positions.

While all final models effectively captured the signal contained in the 4 datasets (Figure 6—figure supplement 1), this signal is fractionated differently in the different landscapes: ppluGFP2 and amacGFP, the least epistatic genes, are fairly well recapitulated by linear models (no interactions); avGFP required the addition of a sigmoid node and cgreGFP required the addition of a subnetwork to fix the linear estimate and yield a correlation that was as good as the models describing other proteins (Figure 6—figure supplement 1). This mirrors the amount of epistasis observed in the different landscapes, which is mostly negative epistasis (Figure 3—figure supplement 1). Furthermore, even though the correlation of the final models is similarly good for all four genes, it is the fragile, highly epistatic cgreGFP which yielded the best predictions. Although the inner workings of neural nets remain a black box, our data does seem to suggest that the epistatic signal of cgreGFP is illustrated well enough in the data for the model to capture it and learn about negative epistatic interactions.

We have added a new Figure 7—figure supplement 1 to highlight the successful incorporation of sometimes-deleterious mutations into functional predictions, which suggests that the model was able to identify and avoid unfavourable interactions.

Also, it was not clear to me that they only used amino acid mutations that were observed in either phylogeny or their deep mutational scanning dataset (for each GFP)? Or they went to other amino acids that they experimentally did not observe.

We used amino acid states observed in our libraries (our ML approach cannot describe mutations not present within the training set) but that were, whenever possible, not observed in phylogeny. We now mention this in the main text and refer the reader to the dataset where this information can be found for each specific genotype we tested.

Reviewer #3 (Recommendations for the authors):In Figure 3, the description of panels A and B appears to be swapped with respect to the Y axis labels.

Thank you, this is now corrected.

The results shown in Figure S5A are unexpected. Taking amacGFP as an example, the graph shows that genotypes with two large-effect mutations, shown in the bottom left corner of this graph, typically show no epistasis (yellow points). According to equation 1, this indicates that the fluorescence of such genotypes should be equal to the sum of effects of the two mutations, i.e. close to -2.4 (-1.2 + -1.2). However, fluorescence = -2.4 appears to be lower than the dynamic range of fluorescence for amacGFP (see for example Figure 1C, where the fluorescence effect of all genotypes appears to be in the range from 0 to -1.5). Can you clarify this? Perhaps a non-additive model of epistasis was used in this figure?

The reviewer is correct that the standard calculation of “expected” fluorescence is simply the addition of all individual mutation effects, however, the possible output values of this calculation are bounded by the lower and upper limits of what was experimentally measurable (i.e., cannot be lower than the minimum, non-fluorescent values, nor higher than the highest measured fluorescence). While the X and Y axes indicate the individual effects of single mutations, their expected joint effect cannot violate these upper and lower bounds. A single mutation with an effect of -1.2 already results in a non-functional protein, and the addition of a second -1.2 mutation cannot decrease fluorescence beyond this already-attained minimum. For this reason, negative epistasis can never be detected between two such highly individually deleterious mutations, because the non-epistatic expectation is already non-functional; dots in this region will therefore always be yellow except in cases of positive epistasis -- when individually deleterious mutations are not deleterious together -- however, positive epistasis was rare in our data. We have updated the figure legend to clarify this point.

Figure S9 legend: amacGFP:V14L should be V12L.

Thank you, corrected.

Figure S9: why were ddG predictions excluded for proline and glycine mutations?

Mutations from/to proline and glycine tend to alter the backbone conformation and require specific handling, for example, an intensive backbone sampling. However, more aggressive remodelling was shown to decrease the ability of the used method to recapitulate mutant structure and can have correspondingly negative impact on ddG prediction for other amino acids. We decided to exclude this small subset of mutations instead of combining data obtained using different methods.

It would be good to discuss the generality of these results. The fitness landscapes of GFP homologues are bimodal, with mutations tending to have either very little effect, or to cause large reduction of fluorescence. Is that a common property of fitness landscapes, or is it specific to GFP, or to the experimental technique used to measure fitness?

To some degree, this issue has recently been reviewed (Kemble et al., Evolutionary Applications, 2019). We also talk about this in the third paragraph of our introduction, but there we are using the language typically used to describe fitness landscape rather than protein structure (but essentially the point is the same). However, the real issue is that there are so few examples of extensive fitness landscape with multiple mutations across an entire protein that we feel it is not possible yet to make generalisations based on comparisons between proteins. So far, we count three such characterizations of fitness landscapes: for Β-lactamase, Hsp90 and GFP protein-coding genes, all three are globular proteins. Would the fitness landscape be different in a membrane protein? In a disordered protein? All of these are interesting questions that we feel require solid experimental data to address. Taken together, we feel that such a generalisation at this point would be premature – the field needs another decade of data gathering. We suspect that the bimodality will be a common feature, but we hardly have the data to support this argument.

Can the data be used to discriminate between second and higher-order epistasis?

Yes, it may. However, we were not interested in this particular analysis. As discussed in the previous question the GFP fitness landscape is generally “bimodal”, the protein is either fluorescent or not. For such fitness functions epistasis is expected to be of maximal order – this is a mathematical feature of phase transition-like functions that was discussed in Pokusaeva et al., PLoS Gen, 2019. Therefore, some higher-order epistasis is likely to be detected in GFPs with a sharp fitness peak, but showing that this is the case, in our opinion, does not offer mechanistic or conceptual understanding of the corresponding fitness landscape. Generally speaking, the sheer volume of the data we produced is such that many other analyses may be performed. We hope that the research community will take advantage of the availability of these data and ask questions we did not think of asking.